# The research and development process for multiscale models of infectious disease systems

**Winston Garira** [ORCID] *

Modelling Health and Environmental Linkages Research Group (MHELRG), Department of Mathematics and Applied Mathematics, University of Venda, Thohoyandou, South Africa

* winston.garira@univen.ac.za, wgarira@gmail.com

**Data Availability Statement:** This article has no data.

**Funding:** The author acknowledges with thanks financial support from South Africa National

## Abstract

Multiscale modelling of infectious disease systems falls within the domain of complexity science—the study of complex systems. However, what should be made clear is that current progress in multiscale modelling of infectious disease dynamics is still as yet insufficient to present it as a mature sub-discipline of complexity science. In this article we present a methodology for development of multiscale models of infectious disease systems. This methodology is a set of partially ordered research and development activities that result in multiscale models of infectious disease systems built from different scientific approaches. Therefore, the conclusive result of this article is a methodology to design multiscale models of infectious diseases. Although this research and development process for multiscale models cannot be claimed to be unique and final, it constitutes a good starting point, which may be found useful as a basis for further refinement in the discourse for multiscale modelling of infectious disease dynamics.

## Author summary

Complex systems such as infectious disease systems are inherently multilevel and multiscale systems. The study of such complex systems is called complexity science. In this article we present a methodology to design multiscale models of infectious disease systems from a complex systems perspective. Within this perspective we define complexity science as the study of the interconnected relationships of the levels and scales of organization of a complex system. We therefore, define the degree of complexity of a complex system as the number of levels and scales of organization of the complex system needed to describe it. In this work we first present a common multiscale vision of the multilevel and multiscale structure of infectious disease systems as complex systems in which the levels and scales of organization of an infectious disease system interact through different self-sustained multiscale cycles/loops (primary multiscale loops, or secondary multiscale loops, or tertiary multiscale loops) formed at different levels of organization of an infectious disease system due to ongoing reciprocal influence between the microscale and the macroscale. Guided by this multiscale vision, we propose a four-stage research and development process that

Research Foundation (NRF) Grant No. IPRR (UID 81235). The funders had no role in study design, data collection and analysis, decision to publish, or preparation of the manuscript.

**Competing interests:** The author declares no competing non-financial/financial interests.

result in multiscale models of infectious disease systems built from different scientific approaches.

## Introduction

A common feature of complex systems is that they are multilevel and multiscale systems. The increasing ability to exhaustively study complex systems such as physical systems, infectious disease systems, food systems, energy systems, water systems, biological systems, chemical systems, artificial systems, and many more, in terms of their levels of organization and their associated scales of observation has raised hopes that this would lead to a systems level description of complex systems using multiscale modelling methods. Multiscale modelling is an emerging scientific method for exploring complex systems. In multiscale modelling of complex systems, there is an appreciation of the complexity of a system arising from its multilevel, multiscale and interconnected relationships occurring within levels of organization and scales of the complex system. In this article, we identify infectious disease systems as one of the complex systems facing major roadblocks due to multiscale needs and formulate a research and development process for multiscale models of infectious disease systems. The conclusive result of this article is a methodology to design multiscale models of infectious diseases. The lack of standardization among scientists using multiscale modelling of infectious diseases research makes it very difficult to achieve consensus in the best methods to create and share these models among the scientific community.

While there has been significant progress in the understanding of the complexity of infectious disease systems, that progress has been limited by a number of diverse challenges that must be overcome [1] in order to realize the full potential of multiscale modelling in characterizing the complexity of infectious disease systems. The article [1], identified ten of the most significant challenges that stand in the way of future advances in multiscale modelling of infectious disease systems. These challenges are problems that have never been solved in a holistic manner before and require collaborative research among scientists with different skills to be fully resolved. A great challenge before us is that how can we use multiscale modelling as a tool to break barriers among scientists with different skills and provide a venue for collaborative research among these scientists to synthesize knowledge about complex systems in a way that establishes multiscale modelling as an indispensable tool for complexity science? Multiscale modelling of infectious disease systems would benefit from interdisciplinary contributions, particularly from mathematical modelling, epidemiology, microbiology, environmental science, population biology, ecology, immunology, and public health research. Efforts to ensure an interdisciplinary approach to multiscale modelling of infectious disease systems still face challenges because of the lack of a common process model that can give a common multiscale vision of the research and development process and enable scientists with different skills to come together to drive multiscale modelling of infectious disease systems. In order to ensure an interdisciplinary approach to multiscale modelling of infectious disease systems in which scientists with different skills collaborate, there is need to develop a shared research and development process for multiscale models of infectious disease systems. Such a shared research and development process for multiscale models of infectious disease systems will give a common multiscale vision to the different scientists of the multilevel and multiscale structure of infectious disease systems, and the inherent feed-backs that exist between the levels and scales of such complex systems. The need for multiscale modelling as opposed to single scale modelling of infectious disease systems comes from:

a. *The need to study infectious disease processes at the scale at which they occur*: Some disease processes are most easily observed and have their greatest impact at a particular scale. If we study a disease process at a different scale from its characteristic scale, we may not detect its actual dynamics, but instead detect/identify its emergent behaviour. For example, the inability to distinguish between local infections and imported infections in the study of infectious disease systems is because of using single scale modelling instead of multiscale modelling.

b. *The need to reduce errors in modelling infectious disease systems*: If a disease process is studied at inappropriate scale than its characteristic scale, there is the possibility of making errors in conclusions of two types due to scale mismatch between the scale at which disease processes occur and the scale at which decisions on them are made. The first type of error arises when conclusions are made at macroscale based on studies from microscale. For example, conclusions are made on between-host scale disease dynamics based on analysis of within-host scale model. The second type of error arises when conclusions are made at microscale based on studies from macroscale. For example, conclusions are made on within-host scale disease dynamics based on analysis of between-host scale model.

c. *The need to incorporate more detail in modelling of infectious disease systems*: Disease dynamics is generally a multiscale phenomena. This implies that infectious disease processes that operate at a particular scale are typically related to disease processes at other scales as well. Therefore, the relationship between different infectious disease processes are often too complex to be fully understood when studied at a single scale of observation. Studies at other scales are often needed to understand the implications of changes at any given scale. Understanding cross-scale influence is key to characterizing the complexity of infectious disease systems. As a result, studies of infectious disease systems conducted at multiple scales provide a fuller characterization of infectious disease dynamics than do those conducted at single scales and tend to explain more of the observed variation.

d. *The need to study factors that influence disease dynamics at scales at which they have greatest impact*: The impact of many factors that influence disease dynamics such as immune response, health interventions and environmental change are typically most strongly expressed, most easily observed, or have their dominant drivers or consequences at particular spatial and temporal scales of an infectious disease system.

In this article, we present a process model for multiscale modelling of infectious disease systems from a perspective of complexity science. We first present key features that help us to build a common multiscale vision of the multilevel and multiscale structure of infectious disease systems in section. Guided by this multiscale vision, the research and development process for multiscale models of infectious disease systems is presented. We conclude the paper with some remarks.

## Methods

Before we can present the process model for multiscale modelling of infectious disease systems, there is need to give a common multiscale vision of their overall multilevel and multiscale structure. This helps to avoid the use of arbitrarily defined levels and scales of an infectious disease system which makes it difficult to establish collaboration among scientists with different skills in the research and development process for multiscale models of infectious disease

systems. Collaboration among scientists with different skills in the research and development process for multiscale models of infectious disease systems would be possible if there exists a common understanding about the hierarchy of components which constitute an infectious disease system starting with its sub-systems. Then the sub-systems are decomposed into levels. Finally the levels are decomposed into scales. Without this common conceptual understanding of the overall multilevel and multiscale structure of an infectious disease system, any defined multiscale modelling study will likely lack a common multiscale vision to facilitate collaboration among scientists from different disciplines (biology, epidemiology, pharmacology, microbiology, public health, immunology, medicine, veterinary science, mathematical modelling, ecology, statistical modelling, environmental science, etc.). In the following sub-sections, we present key features that help us to build a common multiscale vision of the overall multilevel and multiscale structure of infectious disease systems. These features have not been discussed in the past with sufficient detail to enable us to present an interdisciplinary multiscale vision of the research and development process for multiscale models of infectious disease systems.

## An infectious disease system is a complex system made up of three sub-systems

An infectious disease system is a complex system composed of three main interacting sub-systems which are the environmental sub-system, the pathogen sub-system and the host sub-system. This complex nature of infectious disease systems makes their aggregate dynamics non-linear and being characterized by the following properties.

a. *Emergence*—so that patterns emerge from the interaction of an infectious disease system's sub-systems (host sub-system, pathogen sub-system, and environmental sub-system), e.g. colonization of host sub-system by the pathogen sub-system, establishment of pathogen sub-system within the host sub-system while evading immune response system, transmission of the pathogen sub-system to new host sub-systems, altered host sub-system behaviour, use of host sub-system as both transport and reservoir of pathogen sub-system, etc.

b. *Co-evolution*—so that the host sub-system and the pathogen sub-system impose selection on each other in a dynamic process of ongoing reciprocal change where the pathogen sub-system imposes a selective influence on the host sub-system which respond to the selection, in turn imposing a selective influence on the pathogen sub-system, with this cycle potentially repeated over and over with this process potentially involving traits like pathogen sub-system infectivity, host sub-system resistance to infection and pathogen sub-system host-finding ability.

c. *Self-organization*—so that some overall order which is spontaneous (i.e not requiring control from an external agent) and robust i.e self-reinforcing mechanism that causes infectious disease systems to persist and spread due to formation of self-sustained multiscale cycles/loops (primary multiscale loop, or secondary multiscale loop, or tertiary multiscale loop) arises from the interactions between the sub-systems of an infectious disease system (host sub-system, pathogen sub-system, and environmental sub-system) which result in an infectious disease system being organized into a hierarchical multilevel and multiscale structure. This structural characteristic of infectious disease systems is discussed further in this article. The formation of multiscale cycles/loops at every level of organization of an infectious disease system due to ongoing reciprocal influence between the microscale and the macroscale [2] in disease dynamics is discussed in this article.

 d. *Openness*—so that it may be difficult to determine infectious disease system boundaries due to local and global exchange of organisms implicated in disease dynamics (pathogen sub-system and host sub-system) among the levels and scales of an infectious disease system—which is also the basis of the linkage between the levels and scales of an infectious disease system. This functional role/characteristic of the pathogen sub-system and host sub-system in linking the levels and scales of an infectious disease system is discussed further in this article.

 e. *A history*—so that the past helps to shape the present behaviour of an infectious disease system e.g. due to development of partial immunity due to prior exposure to the disease system [3] or development of herd immunity due to prior exposure to vaccination.

This complex nature of infectious disease systems implies that any infectious disease system we want to study in order to control, eliminate or eradicate brings together biological, medical, epidemiological, immunological, clinical, public health issues, and their functional context with multiple interactions and consequences for the health of humans, animals and plants.

## An infectious disease system is a structurally organized complex system

There is need to understand how complex systems are organized, independently of whether those complex systems are physical systems, infectious disease systems, chemical systems, environmental systems, social systems, economic systems, artificial systems or biological systems. The organization of complex systems and their sub-systems can be categorized into two main groups as follows.

 a. *Structurally organized complex systems*: Infectious disease systems are one example of structurally organized complex systems. We consider complex systems or their sub-systems to be structurally organized if they can be resolved into one or more hierarchical levels of organization such that each level of organization has two limiting scales: a microscale and a macroscale. One of the defining properties of such structurally organized complex systems and their sub-systems is that the hierarchical levels of organization exhibit clear spatial and temporal hierarchy, so that the hierarchical levels of organization are multiscale in both space and time (i.e. the scales vary concurrently in both space and time). This implies that along the hierarchy of levels of organization of such structurally organized complex systems or their sub-systems, the temporal and spatial scales of the microscale and macroscale in which processes interact are correlated in that as the spatial scale of the process increases, so does the temporal scale over which the same process takes place. For such complex systems a level is different from a scale. The articles [1, 2] were the first to provide a formal expression of this concept in the context of infectious disease systems in which the correlated scales (microscale and macroscale) of spatial and temporal variability along the hierarchy of levels of organization of an infectious disease system from the cell level (which is the lowest level in the hierarchy) to the macroecosystem level (which is the highest level in the hierarchy) are graphically portrayed and described. In this article we consider the research and development process for multiscale models of structurally organized complex systems with particular reference to infectious disease systems.

 b. *Functionally organized complex systems*: The six main functionally organized complex systems which influence infectious disease dynamics are (i) economic system, (ii) evolutionary system, (iii) social system, (iv) environmental change system, (v) health interventions system, and (vi) immune response system. We consider complex systems to be

functionally organized if they can be resolved into more than one level of organization such that each level of organization has one limiting scale with each of the scales having a temporal scale only. Therefore such complex systems or their sub-systems are multi-level in time with the levels exhibiting a clear temporal hierarchy but without a corresponding spatial hierarchy or the spatial hierarchy is not considered in the complex system's dynamics. Further, it is not easy to determine the spatial boundaries of functionally organized complex systems. For functionally organized complex systems or their sub-systems a level is the same as a scale and scales/levels exhibit a clear temporal hierarchy only. We will discuss further the six main functionally organized complex systems which influence infectious disease dynamics in this article. However, the research and development process for multiscale models of functionally organized complex systems is beyond the scope of this article.

There are ways of determining whether infectious disease systems or their sub-systems are either functionally or structurally organized. We can determine the nature of organization of sub-systems by considering their role in disease dynamics. The three sub-systems (host sub-system, pathogen sub-system, and environmental sub-system) can play two main roles in disease dynamics. A sub-system can play the role of reservoir of infective pathogen (i.e. as pathogen habitat or as a host) or it can play the role of linking levels and scales of an infectious disease system. If a sub-system plays the role of linking levels and scales of an infectious disease system, then it is considered as a functionally organized sub-system. If it plays the role of reservoir of infective pathogen (i.e. as pathogen habitat or as a host), then it is considered as structurally organized sub-system. Therefore, the environmental sub-system is always a structurally organized sub-system because it only plays the role of pathogen habitat without any role in linking the levels and scales of an infectious disease system. However, at lower levels of organization of an infectious disease system the host sub-system is considered as a structurally organized sub-system because it plays the role of pathogen habitat. At higher levels of organization of infectious disease system, the host sub-system can be considered as a functionally organized sub-system because it plays the role of linking levels and scales of an infectious disease system. But, because the environmental sub-system's levels of organization at these higher levels of organization of an infectious disease system are also demarcated into administrative units of the host sub-system, that is, the various levels of organization of the community (local, national, regional, etc.), then the host sub-system is overall considered as a structurally organized complex sub-system. The pathogen sub-system is usually considered as a functionally organized sub-system at all levels of organization of an infectious disease system because its usual role is to link the levels and scales of an infectious disease system. However, within this sub-system, some pathogen types such as the bacterial pathogen can be considered as a structurally organized sub-system in circumstances when it becomes pathogen habitat, i.e. when it is considered as habitat for mobile genetic elements such as phage or plasmids [4]. The resolution of each of the sub-systems of an infectious disease system into different hierarchical levels of organization is considered in the sub-section that follows. By considering the organization of sub-systems of a complex system (i.e. whether they are structurally organized or functionally organized), we can also be able to determine the organization of the overall complex system. If all the sub-systems of a complex system are functionally organized, then the resulting overall complex system is a functionally organized complex system. However, if one or more of the sub-systems of a complex system are structurally organized, while the rest of them are functionally organized, then the resulting overall complex system is a structurally organized complex system. Infectious disease systems are always structurally organized complex systems because the environmental sub-system is always a structurally organized sub-system. In

general, for structurally organized complex systems, we are interested in how processes at mutually exclusive scales (in both space and time) influence the dynamics of the complex system. For functionally organized complex systems, we are interested in how mechanisms that usually do not operate at mutually exclusive spatial scales influence the dynamics of the complex system at mutually exclusive temporal scales. In the following sub-section, we discuss in detail the levels and scales of infectious disease systems.

## The sub-systems of an infectious disease system can be resolved into levels and scales

In our previous work, we have progressively refined the framework for the multilevel and multiscale structure of infectious diseases (as new insights/knowledge emerged) in three different articles [1, 2, 5]. The frameworks in [1, 2, 5] differ from the framework in the current article in the following aspects. In [5], only a few representative levels of organization of an infectious disease system and associated scales were included in the framework. The framework in [5] was refined in [1] by including all the main levels of organization of an infectious disease system without making distinction between the microecosystem level and the macroecosystem level. Further, the framework in [1] was refined in [2], by making distinction between the microecosystem level and the macroecosystem level and emphasizing the role of these levels in the multiscale modelling of the ecology and evolution of infectious disease systems. However, in [2] we did not include information on how the levels are related to the different sub-systems of an infectious disease system and how the levels and scales are linked through exchange of organisms implicated in the transmission of an infectious disease system. In the current article, the framework in [2] is further refined by including details of how the levels are derived from the various sub-systems of an infectious disease system. We also include details of how the levels and scales are linked through exchange of organisms implicated in the transmission of an infectious disease system. The three sub-systems of an infectious disease system are organized into the following levels and scales:

a. *The host sub-system*: A type of structurally organized complex sub-system of an infectious disease system which consists of seven main levels of organization with each level having two limiting scales as follows. (i) The cell level—which has the within-cell scale and the between-cell scale. (ii) The tissue level—which has the within-tissue scale and the between-tissue scale. (iii) The organ level—which has the within-organ scale and the between-organ scale. (iv) The microecosystem level—which has the within-microecosystem scale and the between-microecosystem scale. (v) The host/organism level—which has the within-host scale and the between-host scale. (vi) The community level—which has the within-community scale and the between-community scale. (vii) The macroecosystem level—which has the within-macroecosystem scale and the between-macroecosystem scale.

b. *The environmental sub-system*: A type of structurally organized complex sub-system of an infectious disease system which consists of two main levels of organization which are the microenvironmental level (which we alternatively call the inside-host environmental level) and the macroenvironmental level (which we alternatively call the outside-host environmental level). The outside-host environmental level or the community level has three sub-levels which are as follows. (i) The local community level—which has the within-local community scale and the between-local community scale as the microscale and the macroscale. (ii) The territorial community level, that is, national/country level—which has the within-nation/country scale and the between-nation/country scale as the

microscale and the macroscale. (iii) The regional community level—which has the within-region scale and the between-region scale as the microscale and the macroscale. Equally, the inside-host environmental level has four main sub-levels which are as follows. (i) The cell level—which has the within-cell scale and the between-cell scale as the microscale and the macroscale. (ii) The tissue level—which has the within-tissue scale and the between-tissue scale as the microscale and the macroscale. (iii) The organ level —which has the within-organ scale and the between-organ scale as the microscale and the macroscale. (iv) The microecosystem level—which has the within-microecosystem scale and the between-microecosystem scale as the microscale and the macroscale. The environmental sub-system defines the spatial scales over which the interactions of the other two complex sub-systems that constitute an infectious disease system (the pathogen sub-system and the host sub-system) take place (i.e. where host-pathogen interactions take place).

c. *The pathogen sub-system*: A type of functionally organized complex sub-system of an infectious disease system which consists of two main levels of organization, with each level being the same as a scale. The two main levels/scales of organization of the pathogen sub-system are (i) the single pathogen species/strain level/scale and (ii) the multiple pathogen species/strains level/scale. In the overall infectious disease system, the pathogen sub-system functions to link the levels and scales of an infectious disease system. The functional role of the pathogen sub-system in linking the scales and levels of an infectious disease system is discussed further in in this article. However, in circumstances where there is horizontal gene transfer, e.g., via phage or plasmids, which can occur between pathogen species that occupy the same niche/host and between strains of the same species during co-infection, then the pathogen sub-system is considered as a structurally organized sub-system. This horizontal gene transfer is especially important for the spread of antibiotic resistance in pathogen populations, and for vaccine escape, such as that which occurred for Pneumococcus [6].

Fig 1 shows a schematic diagram of the resolution of an infectious disease system into subsystems first (host sub-system, environmental sub-system, and pathogen sub-system), then the resolution of each sub-system into hierarchical levels of organization, and finally the resolution of each hierarchical level into scales. However, we did not illustrate the structural role of the pathogen sub-system in Fig 1 which occurs during horizontal gene transfer, e.g., via phage or plasmids in order keep it simple.

## There are two types of levels of organization of an infectious disease system which are also the basis for determining the boundaries of scales

For structurally organized complex systems, such as infectious disease systems, we identify two different types of levels of organization which are:

a. *A level of multiscale analysis*: This is the level of organization of an infectious disease system that frames what is to be analyzed or is the level of organization of an infectious disease system that is studied as a whole, within which the scales of analysis exist. Thus, a level of multiscale analysis is the level of organization of an infectious disease system the multiscale modeler would like to say something about and make conclusions about it. For infectious disease systems, the levels of organization of an infectious disease system from the tissue level to the macroecosystem level are potentially levels of multiscale analysis. In multiscale modelling of infectious diseases the microscale associated with each of these levels of multiscale analysis is used as a scale of analysis—which in turn is used to

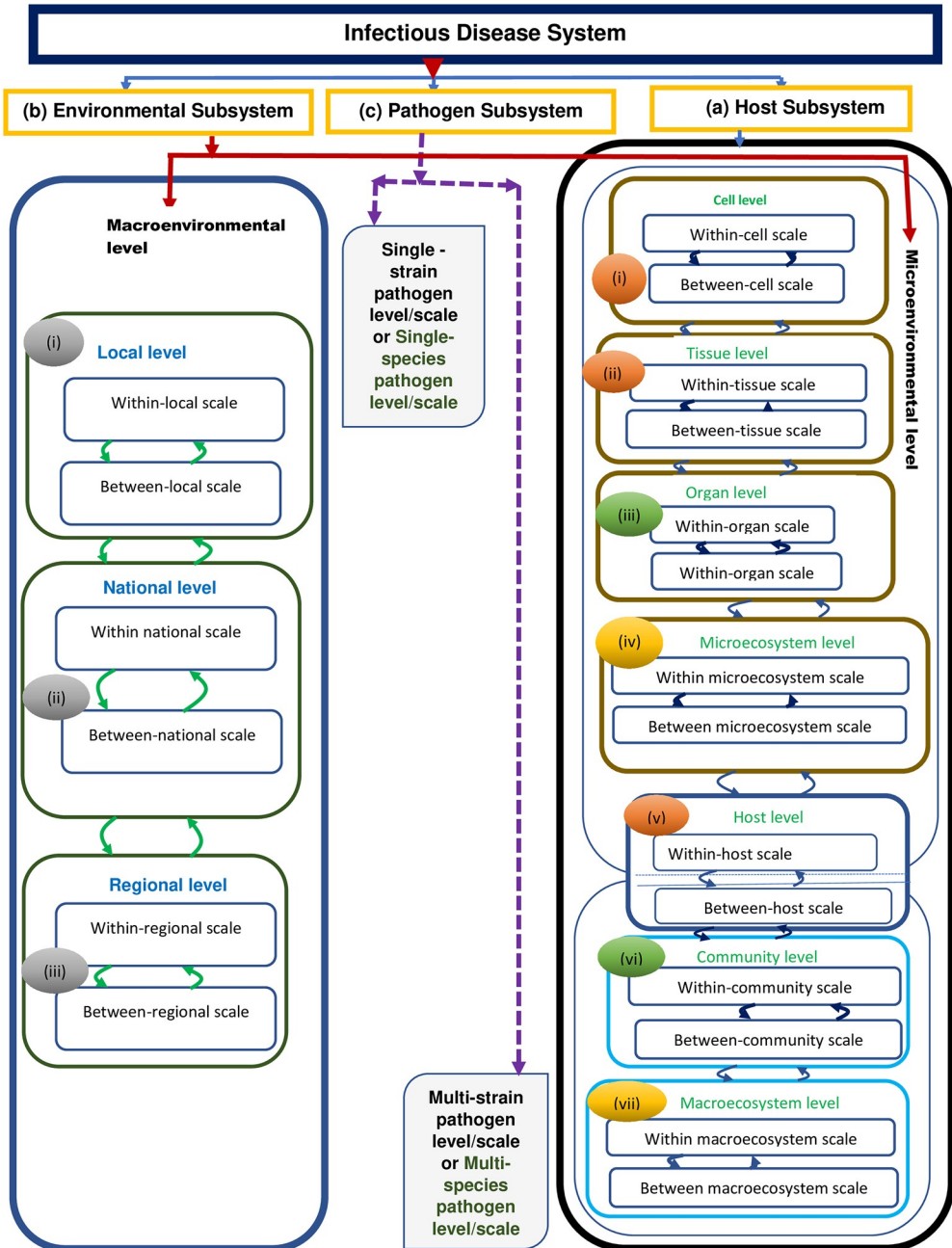

**Fig 1. A schematic diagram of the resolution of an infectious disease system into the three sub-systems.** (a) The host sub-system which consists of seven main levels (i) the cell level, (ii) the tissue level, (iii) the organ level, (iv) the microecosystem level, (v) the host/organism level, (vi) the community level, and (vii) the macroecosystem level. (b) The environmental sub-system which consists of two main levels of organization which are the inside-host environmental level and the outside-host environmental level (or community level). The outside-host environmental level (or community level) has three sub-levels which are (i) the local community level, (ii) the territorial community level, and (iii) the regional community level. (c) The pathogen sub-system which in its functional role consists of two main levels of organization, with each level being the same as a scale which are (i) the single pathogen species/strain level/scale and (ii) the multiple pathogen species/strains level/scale. However, in circumstances where there is horizontal gene transfer, the pathogen sub-system would consist of (i) the single pathogen species/strain level and (ii) the multiple pathogen species/strains level. In this Figure, we did not illustrate this structural role of the pathogen sub-system in order keep it simple.

determine boundaries of scales of observation associated with each level of multiscale observation. An infectious disease system has seven main scales of analysis which are (i) within-tissue scale, (ii) within-organ scale, (iii) within-microecosystem scale, (iv) within-host scale, (v) within community scale, (vi) within-macroecosystem scale, (vii) within-whole world scale. For these scales of analysis, direct measurement of infectious disease dynamics is usually done in terms of a level of multiscale observation.

b. *A level of multiscale observation*: This is the level of organization of an infectious disease system that you actually observe, measure or collect data in trying to learn/study something about the level of multiscale analysis or the scale of analysis. A level of multiscale observation is an indirect way of measuring a level of multiscale analysis or a scale of analysis. Therefore, a level of multiscale observation is simply a question of looking for proxies that can stand for the unmeasurable level of multiscale analysis or scale of analysis. For infectious disease systems, all the seven levels of organization of an infectious disease system from the cell level to the macroecosystem level are potentially levels of multiscale observation.

Together, a level of multiscale analysis (and its scales of analysis) and a level of multiscale observation (and its scales of observation) help to define a specific multiscale problem to be investigated. For any given multiscale modelling study, the level of multiscale analysis may not be the same as the level of multiscale observation. An important point to note here is that a level of multiscale observation can never be a higher level of organization of an infectious disease system than its corresponding level of multiscale analysis. From the level of multiscale analysis and the level of multiscale observation we can be able to establish scales of analysis and scales of observation and their boundaries. In general, for structurally organized complex systems, each level of organization (level of multiscale analysis or level of multiscale observation) has two limiting scales which are the microscale and the macroscale. However, for functionally organized complex systems, each level of organization (level of multiscale analysis or level of multiscale observation) has one scale of analysis or scale of observation. Since infectious disease systems are structurally organized complex systems, each level of organization (level of multiscale analysis or level of multiscale observation) has two limiting scales which are the microscale and the macroscale. Further, from the level of multiscale observation and the scale of analysis, we can define the boundaries of the microscale and the macroscale of the level of multiscale observation used in the development of a multiscale model. There is no unique way of choosing the boundaries of scales of observation (the microscale and the macroscale) for each level of multiscale observation used in the development of a multiscale model. The choice of boundaries of scales of observation of a complex system depends on the scale of analysis used in the multiscale study. For each level of multiscale observation of an infectious disease system, the boundary of the microscale is fixed. However, the boundary of the macroscale is not fixed, and is determined by the boundary of the scale of analysis. This is because, for an infectious disease system, a scale of analysis can be described in terms of a level of multiscale observation. We give four examples to illustrate the determination of the boundaries of the microscale and macroscale for specific levels of multiscale observation when the scale of analysis is given as follows:

a. *Boundaries of the within-cell scale and between-cell scale when the within-tissue scale is the scale of analysis*: If the aim of the study is focused on disease dynamics in the tissue micro-environment, that is, the within-tissue scale (e.g. within-granuloma scale) is the scale of analysis, then the boundary of the within-cell scale is the cell wall while the boundary of the within-tissue scale (i.e. the fibroblastic rim punctuated by lymphocytes

that encloses a granuloma) becomes the boundary of the between-cell scale. This implies that the within-tissue scale as a scale of analysis can be described by the cell level as the level of multiscale observation (for which the within-cell scale and the between-cell scale are the microscale and the macroscale with the within-tissue scale boundary as the boundary for the between-cell scale).

b. *Boundaries of the within-cell scale and between-cell scale when the within-organ scale is the scale of analysis*: If the scale of analysis is the within-organ scale, then the boundary of the within-cell scale is also the cell wall while the boundary of the within-organ scale (i.e. a sheath that encloses an organ such as the Pericardium, a double-walled sac containing the heart and the roots of the great vessels or the thin double-layered membrane known as the Pleural membrane that encloses the lungs) becomes the boundary of the between-cell scale. This implies that the within-organ scale as a scale of analysis can be described by the cell level as the level of multiscale observation (for which the within-cell scale and the between-cell scale are the microscale and the macroscale but with the within-organ scale boundary as the boundary for the between-cell scale).

c. *Boundaries of the within-cell scale and between-cell scale when the within-host scale is the scale of analysis*: If the scale of analysis is the within-host scale, then similarly, the boundary of the within-cell scale is the cell wall while the boundary of the within-host scale (i.e. the skin for most vertebrate hosts or a hard outer skeleton in invertebrate hosts as in most mollusks, crustaceans, and insects) becomes the boundary of the between-cell scale. This implies that the within-host scale as a scale of analysis can be described by the cell level as the level of multiscale observation (for which the within-cell scale and the between-cell scale are the microscale and the macroscale but with the within-host scale boundary as the boundary for the between-cell scale).

d. *Boundaries of the within-host scale and between-host scale when the within-community scale is the scale of analysis*: If the scale of analysis is the within-community scale, then the boundary of the within-host scale is still the skin for most vertebrate hosts or a hard outer skeleton in invertebrate hosts as in most mollusks, crustaceans, and insects while the boundary of the within-community scale ((i) within-local community scale whose boundaries are administrative boundaries which are legally documented and attributed jurisdictional boundaries based on administrative system of a country, or (ii) within-country scale, or (iii) within-region scale, or (iv) within-whole world scale) becomes the boundary of the between-host scale. This implies that the within-community scale as a scale of analysis can be described by the host level as the level of multiscale observation (for which the within-host scale and the between-host scale are the microscale and the macroscale but with the within-community scale boundary as the boundary for the between-host scale).

Determining the boundaries of the microscale and the macroscale for each level of multiscale observation used in the development of the multiscale model is an essential part in multiscale modelling of infectious disease systems.

## An infectious disease system has seven main levels of organization

From Fig 1, we notice that the host sub-system has seven main levels of organization. This implies that host-pathogen interactions can play out at any of these seven main levels of organization of the host sub-system. In addition, the environmental sub-system's levels of organization are demarcated into administrative units of the host sub-system at higher levels of its

organization, that is, the various community levels (local, national, regional). Therefore, because of this, we end up with seven main distinct levels of organization of an infectious disease system where host-pathogen interactions can play out which are (i) the cell level, (ii) the tissue level, (iii) the organ level, (iv) the microecosystem level, (v) the host/organism level, (vi) the community level, and (vii) the macroecosystem level. Five different categories of multiscale models of infectious disease systems can be developed, with each of these seven main levels of organization of an infectious disease system as a level of multiscale observation within the seven main scales of analysis ((i) within-tissue scale, (ii) within-organ scale, (iii) within-microecosystem scale, (iv) within-host scale, (v) within community scale, (vi) within-macroecosystem scale, (vii) within-whole world scale). The five main different generic categories of multiscale models of infectious disease systems that can be developed with each of the seven main levels organization of infectious disease systems as a level of multiscale observation are [1, 2, 5]: (i) individual-based multiscale models (IMSMs), (ii) nested multiscale models (NMSMs), (iii) embedded multiscale models (EMSMs), (iv) hybrid multiscale models (HMSMs), and (v) coupled multiscale models (CMSMs) with each of these categories having several different classes of multiscale models. These five categories of multiscale models of infectious disease systems are fully explained in three of our previous papers [1, 2, 5]. In [5] the categories are given including the categorization framework, together with several examples for each category and for each level of organization of an infectious disease. The examples from literature are summarized in Tables for each category. In [1], we summarized the information needed to understand the five categories of multiscale models of infectious disease systems. In [2], we further explained these five categories in terms of the the different forms of reciprocal influence that exist between the the microscale and the macroscale in these different categories of multiscale models. Now, we briefly describe each of the seven main levels of organization of an infectious disease system and the category of multiscale model that can be developed for each of the seven main levels of organization of an infectious disease system ((i) the cell level, (ii) the tissue level, (iii) the organ level, (iv) the microecosystem level, (v) the host/organism level, (vi) the community level, and (vii) the macroecosystem) as a level of multiscale observation in what follows.

a. *The cell level (cL)*: This level of organization has the within-cell scale and between-cell scale as its microscale and macroscale respectively. Different types of target cells can be used as levels of multiscale observation in the development of multiscale models of the dynamics of infectious disease systems such as $CD4^+$ T cells and macrophages (for HIV) or red blood cells and hepatocytes (liver) cells (for malaria) when integrating the within-cell scale and between-cell scale. Multiscale models of infectious disease systems at this level of organization are developed using the cell level (for which the within-cell scale and the between-cell scale are the microscale and the macroscale) as the level of multiscale observation with the (i) within-tissue scale, or (ii) within-organ scale, or (iii) within-host scale as the scale of analysis. Based on the categorization of multiscale models in [1, 2, 5], the multiscale models developed at this level of organization of an infectious disease system fall in categories I, II, III, and IV.

b. *The tissue level (TL)*: This level of organization has the within-tissue scale and between-tissue scale as its microscale and macroscale respectively. The different types of tissues that can be considered in the development of multiscale models of infectious disease systems include granuloma [7] for tuberculosis or microabscess [8] caused by some bacterial infections. Multiscale models of infectious disease systems at this level of organization are developed using the tissue level (for which the within-tissue scale and the between-tissue scale are the microscale and the macroscale) as the level of multiscale

observation with the (i) within-organ scale, or (ii) within-host scale as the scale of analysis. Based on the categorization of multiscale models in [1, 2, 5], the multiscale models developed at this level of organization of an infectious disease system also fall in categories I, II, III, and IV.

c. *The organ level (OL)*: The microscale and macroscale for this level of organization of an infectious disease system are the within-organ scale and the between-organ scale. Multiscale models at this level of organization of an infectious disease system are developed using the organ level (for which the within-organ scale and the between-organ scale are the microscale and the macroscale) as the level of multiscale observation with the within-host scale as the scale of analysis. Some of the organs considered at this level of organization of an infectious disease system are the lung, brain, gut, kidney, muscle, heart, pancreas, stomach, liver, spleen, bone, adrenal, skin, adipose, and blood. Based on the categorization of multiscale models in [1, 2, 5], the multiscale models developed at this level of organization of an infectious disease system fall in category V.

d. *The microecosystem level (mL)*: This is the level of organization of an infectious disease system at which multiscale models of the ecology and evolution of an infectious disease system are developed. The microscale and macroscale for this level of organization of an infectious disease system are the within-microecosystem scale and the between-microecosystem scale respectively. The microecosystem level here consists of three sub-levels. (i) The microecosystem level I—for which the cell level microenvironment as a level of multiscale observation at the within-tissue scale as a scale of analysis is considered as an ecosystem, i.e. infected with multiple pathogen species/strains. The microecosystem level I has the within-microecosystem scale I and between-microecosystem scale I as the microscale and macroscale, respectively. (ii) The microecosystem level II—for which the tissue level microenvironment as a level of multiscale observation at the within-organ scale as a scale of analysis is considered as an ecosystem, i.e. infected with multiple pathogen species/strains. The microecosystem level II has the within-microecosystem scale II and between-microecosystem scale II as the microscale and macroscale, respectively. (iii) The microecosystem level III—for which the organ level microenvironment as a level of multiscale observation at the within-host scale as a scale of analysis is considered as an ecosystem, i.e. infected with multiple pathogen species/strains. The microecosystem level III has the within-microecosystem scale III and between-microecosystem scale III as the microscale and macroscale, respectively. Because of the multiple pathogen species/strains interactions at this microecosystem level (i.e. microecosystem level I, microecosystem level II, microecosystem level III) of organization of an infectious disease system, ecological processes/interactions influence infectious disease dynamics at this level of organization which include the competitive pathogen species/strains interactions and the mutualistic interactions between the multiple pathogen species/strains. Based on the categorization of multiscale models in [1, 2, 5], the multiscale models developed at this level of organization of an infectious disease system fall in category V.

e. *The host/organism level (HL)*: This level of organization has the within-host scale and between-host scale as its microscale and macroscale respectively. Multiscale models of infectious disease systems at this level of organization are developed using the host level (for which the within-host scale and the between-host scale are the microscale and the macroscale) as the level of multiscale observation with the within-community scale (e.g. (i) within-local community scale, or (ii) within-territorial community scale, or (iii) within-regional scale, or (iv) within-whole world scale) as the scale of analysis. Based on

the categorization of multiscale models in [1, 2, 5], the multiscale models developed at this level of organization of an infectious disease system fall in categories I, II, III, and IV.

f. *The community level (CL)*: The microscale and macroscale for this level of organization of an infectious disease system are the within-community scale and the between-community scale. Multiscale models of infectious disease systems at this level of organization are developed using the community level as the level of multiscale observation with the within-community scale (e.g. (i) within-local community scale, or (ii) within-territorial community scale, or (iii) within-regional scale, or (iv) within-whole world scale) as the scale of analysis in the context of single pathogen species/strain. The three main sub-levels of the community level are: (i) local community level (village level, district level, town level, city level, province level, etc.) or other local administrative units (workplace, farm, game reserve, school, university, hospital, etc.), (ii) territorial community level (i.e. nation/country level) and (iii) regional community level (e.g. the six World Health Organization (WHO) regions of the world: African Region, Region of the Americas, South-East Asia Region, European Region, Eastern Mediterranean Region, and Western Pacific Region), which consists of administrative groupings of nations for health purposes. Therefore, the community level consists of three main sub-levels. (i) The community level I—for which the local community level as a level of multiscale observation at the within-country/nation scale or within-region scale or within-whole world scale as the scale of analysis is infected with single pathogen species/strain and/or single host species. The community level I has the within-community scale I and between-community scale I as the microscale and macroscale, respectively. (ii) The community level II—for which the national/country level macroenvironment as the level of multiscale observation at the within-region scale or within-whole world scale as the scale of analysis is infected with single pathogen species/strain and/or single host species. The community level II has the within-community scale II and between-community scale II as the microscale and macroscale, respectively. (iii) The community level III—for which the regional level as a level of multiscale observation at the within-whole world scale as the scale of analysis is infected with single pathogen species/strain and/or single host species. The community level III has the within-community scale III and between-community scale III as the microscale and macroscale, respectively. Based on the categorization of multiscale models in [1, 2, 5], the multiscale models developed at this level of organization of an infectious disease system fall in category V.

g. *The macroecosystem level (ML)*: This is also another level of organization of an infectious disease system at which multiscale models of the ecology and evolution of an infectious disease system are developed. At this level of organization of an infectious disease system, the three main sub-levels of the community level which are (i) the local community level, (ii) national/country level, and (iii) regional level are each considered as an ecosystem, i.e. infected with multiple pathogen species/strains and/or multiple host species. At this level of organization of an infectious disease system, multiscale models are developed with each of these three sub-levels of the of the macroecosystem level (local level, or national level, or regional level) as the level of multiscale observation as follows. (i) The macroecosystem level I—for which the local community level macroenvironment as the level of multiscale observation at the within-country/nation scale or the within-region scale or the within-whole world scale as the scale of analysis is considered as an ecosystem, i.e. infected with multiple pathogen species/strains and/or multiple host species. The macroecosystem level I has the within-macroecosystem scale I and between-

macroecosystem scale I as the microscale and macroscale, respectively. (ii) The macroecosystem level II—for which the national/country level macroenvironment as the level of multiscale observation at the within-region scale or the within-whole world scale as the scale of analysis is considered as an ecosystem, i.e. infected with multiple pathogen species/strains and/or multiple host species. The macroecosystem level II has the within-macroecosystem scale II and between-macroecosystem scale II as the microscale and macroscale, respectively. (iii) The microecosystem level III—for which the regional level macroenvironment as the level of multiscale observation at the within-whole world scale as the scale of analysis is considered as an ecosystem, i.e. infected with multiple pathogen species/strains and/or multiple host species. The macroecosystem level III has the within-microecosystem scale III and between-macroecosystem scale III as the microscale and macroscale, respectively. Because of the multiple pathogen species/strains and multiple host species interactions at this level of organization of an infectious disease system, ecological processes/interactions influence infectious disease dynamics which include predator/prey interactions, competitive pathogen and/or host species interactions and the mutualistic interactions between the multiple pathogen species/strains and/or multiple host species. Based on the categorization of multiscale models in [1, 2, 5], the multiscale models developed at this level of organization of an infectious disease system fall in category V.

To bring some order to the discussion of multiscale models, we propose that each multiscale model be named after the level of multiscale observation of an infectious disease system at which it is developed so that we have seven main different types of multiscale models which are: (i) cell level multiscale models (cL-MSMs), (ii) tissue level multiscale models (TL-MSMs), (iii) organ level multiscale models (OL-MSMs), (iv) microecosystem level multiscale models (mL-MSMs), (v) host level multiscale models (HL-MSMs), (vi) community level multiscale models (CL-MSMs), and (vii) macroecosystem level multiscale models (ML-MSMs). Fig 2 shows a schematic representation of seven main hierarchical levels of organization of infectious disease systems in a space-time diagram which portrays the hierarchical nature of these levels and the positive correlation in the spatial and temporal scales of varying disease processes. The Figure (Fig 2), also shows the four main biological linking mechanisms of these hierarchical levels and their associated scales. The four biological linking mechanisms are represented by circular shapes drawn in dotted lines. The linking of levels and scales of infectious disease systems is discussed further in the following sub-section.

## The levels and scales of an infectious disease system are linked through exchange of organisms implicated in the transmission of an infectious disease system

The most difficult part in multiscale modelling of infectious disease systems is the science devoted to bridging/coupling/linking/integrating the microscale and the macroscale at each level of organization of infectious disease systems, which is also the basis of complexity science. In general, the levels and scales of structurally organized complex system are linked through (i) exchange of forces, (ii) exchange of energy, (iii) exchange of material/matter, or (iv) exchange of information. This is because for structurally organized complex systems, there exists some open scale boundary which divides each level into two adjacent scales (a microscale and a macroscale) which allows for (i) exchange of forces, (ii) exchange of energy, (iii) exchange of material/matter, or (iv) exchange of information between the two adjacent scale (a microscale and a macroscale). However, the scales of functionally organized complex systems are usually linked through some mechanisms which produce some effect at other scales. For

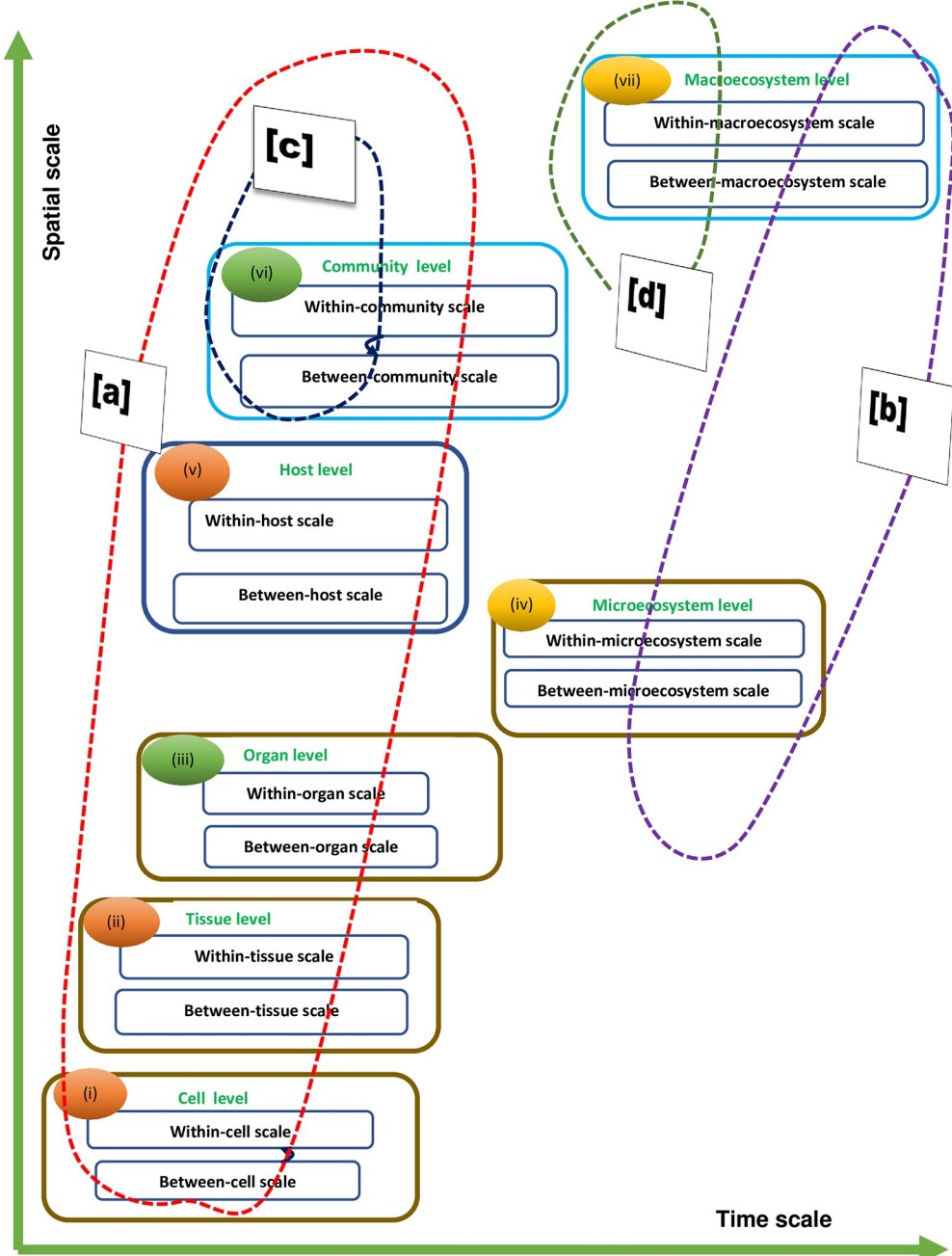

**Fig 2. The schematic diagram of the seven main hierarchical levels of organization of an infectious disease system and the associated scales and the four main biological linking mechanisms of the levels and scales: [a.] linked through exchange of single pathogen species/strain, [b.] linked through exchange of multiple pathogen species/ strains, [c.] linked through exchange of single infected host species/host species products, and [d.] linked through exchange of infected multiple host species/host species products.** The linkage of the scales of an infectious disease system through mobile genetic elements such phage or plasmids is not shown in this Figure in order to keep it simple.

structurally organized complex systems, the challenge is how to represent the linkages (exchange of forces, exchange of energy, exchange of material/matter, exchange of information) in the development of multiscale models. For infectious disease systems, which are also structurally organized complex systems, the levels and scales are linked through exchange of

organisms implicated in the transmission of an infectious disease system which are: (i) the pathogen (virus, prion, helminth, protozoan, bacteria, fungus), (ii) infected hosts (human, animal, vector, plant), (iii) contaminated/infected host products (food, meat, fruits, etc.), and (iv) mobile genetic elements such as phage and plasmids. The exchange of organisms implicated in the dynamics of an infectious disease system involves movement of organisms both locally and globally across levels and scales of an infectious disease system as follows.

a.  *Local exchange of organisms implicated in the dynamics of infectious disease systems*: This exchange of organisms implicated in disease dynamics involves movement of organisms at the scales of three lower levels of organization of an infectious disease system (the cell level, the tissue level, and the host level) and two sub-levels of the microecosystem level (i.e. the microecosystem level I, the microecosystem level II). At each of these lower levels and sub-levels of an infectious disease system (the cell level, the tissue level, the microecosystem level I, the microecosystem level II, and the host level), the only organisms exchanged between scales and levels are either single pathogen species/strain at three lower levels of organization of an infectious disease system (the cell level, the tissue level, host level), or multiple pathogen species/strains at two sub-levels of the microenvironment sub-system (i.e. the microecosystem level I, and the microecosystem level II). This local exchange of organisms implicated in the transmission of an infectious disease system involves a primary multiscale loop at the three lower levels of organization of an infectious disease system (the cell level, the tissue level, and the host level) and a tertiary multiscale loop at the two sub-levels of the of the microecosystem level (the microecosystem level I, and the microecosystem level II). A common feature of these two different multiscale loops (primary multiscale loop and tertiary multiscale loop) at these levels of organization of an infectious disease system is that in both cases it is a replication-transmission multiscale cycle/loop in pathogen dynamics involving four key disease processes which are: (i) infection or super-infection by pathogen (i.e. repeated infection by a pathogen of the same species/strain before the host recovers from prior infection by the same pathogen species/strain), (ii) pathogen replication, (iii) pathogen shedding/excretion, and (iv) pathogen transmission. At these levels and sub-levels of multiscale observation, a replication-transmission multiscale cycle/loop in pathogen dynamics is created because the characteristic scale at which pathogen replication occurs—which is usually the microscale of each of these levels of multiscale observation (i.e. within-cell scale, within-tissue scale, within-microecosystem scale I, within-microecosystem scale II, and within-host scale) and pathogen transmission occurs—which is usually the macroscale of each of these levels of multiscale observation (i.e. between-cell scale, between-tissue scale, between-microecosystem scale I, between-microecosystem scale II, and between-host scale) often do not match. Further, at these levels of organization of an infectious disease system, the macroscale influences the microscale through infection or super-infection (which involves movement of pathogen from macroscale into the microscale) while the microscale influences the macroscale through pathogen shedding/excretion (which also involves movement of pathogen from microscale into the macroscale). Therefore, these multiscale loops involve infection or super-infection and shedding/excretion of the pathogen which introduce a multiscale cycle of influence between pathogen replication at the microscale and pathogen transmission at the macroscale. However, the main distinguishing feature between the primary multiscale loop and the tertiary multiscale loop here is that in a primary multiscale loop only a single pathogen species/strain is exchanged between the scales of an infectious disease system while in a

tertiary multiscale loop multiple pathogen species/strains are exchanged between the scales of an infectious disease system.

b. *Global exchange of organisms implicated in the dynamics of infectious disease systems*: This exchange of organisms implicated in disease dynamics involves movement of organisms at three main levels of organization of an infectious disease system which are the organ level, the community level (the community level I, the community level II, the community level III), and the macroecosystem level (the macroecosystem level I, the macroecosystem level II, the macroecosystem level III) and one sub-level of the microecosystem level (i.e. the microecosystem level III). At these levels and a sub-level of organization of an infectious disease system, the organisms exchanged between scales and levels are either (i) single pathogen species/strain at organ level and community level, or (ii) multiple pathogen species/strains at microecosystem level III and the macroecosystem level, or (iii) single host species/host species products infected/contaminated with single pathogen species/strain at community level, or (iv) multiple host species/host species products infected/contaminated with single pathogen species/strain or multiple pathogen species/strains at macroecosystem level. In the global exchange of pathogen (single pathogen species/strain, or multiple pathogen species/strains) at the organ level or the microecosystem level III, the transport system of hosts (humans, animals, vectors, plants) carries pathogen across these levels of organization of an infectious disease system and their associated scales. For the human and other animal hosts, this involves the circulatory system which encompasses the cardiovascular system and the lymphatic system. For the plant hosts, this involves the vascular system which encompasses the xylem and phloem. This global exchange of pathogen involves movement of pathogen to different anatomical compartments of the host. However, in the global exchange of pathogen for some environmentally transmitted infectious disease systems at the community level and the macroecosystem level, the transport of pathogen happens through aerial means (i.e. through wind/air flow to different geographical areas of the community (local, national, regional, whole world), or through hydrological means (i.e. through water flow) to different levels of organization of the geographical environment (local level, national level, regional level). But, for some environmentally transmitted infectious diseases and directly transmitted infectious diseases, the global exchange of infected human/animal/plant hosts and their contaminated products at the community level and the macroecosystem level happens through travel/migration (for human hosts and animal hosts) or export/import of food (for animal and plant hosts and their products). The global exchange of organisms implicated in the transmission of infectious disease systems at the organ level and the community level involves a secondary multiscale loop while the global exchange of organisms implicated in the transmission of infectious disease systems at the microecosystem level III and the macroecosystem level (the macroecosystem level I, the macroecosystem level II, the macroecosystem level III) involves a tertiary multiscle loop. Unlike the local exchange of organisms implicated in the transmission of infectious disease systems which involves a pathogen replication-transmission multiscale loop (primary multiscale loop or tertiary multiscale loop), the global exchange of organisms implicated in the transmission of infectious disease systems involves an infection persistence-dispersal multiscale loop (secondary multiscale loop or tertiary multiscale loop). The local exchange of mobile genetic elements such as phage and plasmids at pathogen level and their global exchange at other higher levels of organization of an infectious disease system may involve either a primary multiscale loop, or a secondary multiscale loop, or a tertiary multiscale loop.

In general local exchange of organisms implicated in the dynamics of infectious disease systems (which involves either a primary multiscale loop or a tertiary multiscale loop) is responsible for persistence/amplification of infection while global exchange of organisms implicated in the dynamics of infectious disease systems (which involves either a secondary multiscale loop or a tertiary multiscale loop) is responsible for dispersal/spread of infection to different anatomical compartments or organs of the host (e.g. liver, blood, brain, etc. for human and other animal hosts) and to different geographical areas of the world (local, national, regional, etc.).

## There are three main different multiscale loops/cycles which are the basis of a broader scientific theory for multiscale modelling of infectious disease systems

We can identify three main different types of multiscale loops which are the basis for the broader scientific theory for multiscale modelling of infectious disease systems established in [2] which are as follows.

a. *Primary multiscale loop/cycle*: This is a pathogen replication-transmission multiscale loop/cycle which occurs in multiscale dynamics of infectious disease systems. It occurs when the cell level or the tissue level or the host level is used as the level of multiscale observation in the development of multiscale models of infectious disease systems. This multiscale loop involves local exchange of pathogen in the context of single cell species (for the cell level) or single tissue species (for the tissue level) or single host species (for the host level). This primary multiscale loop involves the reciprocal influence between pathogen replication at the microscale of these levels of organization of an infectious disease disease system (i.e. within-cell scale, within-tissue scale, and within-host scale), and pathogen transmission at the macroscale of these levels of organization of an infectious disease disease system (i.e. between-cell scale, between-tissue scale, and between-host scale). In the context of this study, the three levels of multiscale observation (the cell level, the tissue level, the host level) are called primary levels of multiscale observation. At these primary levels of multiscale observation, the organisms exchanged through local exchange are either single pathogen species or single pathogen strain.

b. *Secondary multiscale loop/cycle*: This is an infection persistence-dispersal multiscale loop/cycle which occurs in multiscale dynamics of infectious disease systems. It occurs when the organ level or the community level is used as the level of multiscale observation in the development of multiscale models of infectious disease systems. At the organ level, this secondary multiscale loop exists when the within-host scale is used as a scale of analysis and the organ level is used as the level of multiscale observation. In this case there will be reciprocal influence between persistence of infection at microscale (within-organ scale) and dispersal/spread of infection at macroscale (between-organ scale). This happens when pathogen spreads at between-organ scale (influenced by persistence of pathogen at within-organ scale) to organs with higher levels of susceptibility or persistence, and being re-introduced back after a period of time at within-organ scale (influenced by dispersal/spread of pathogen at between-organ scale). Consider for example HIV/AIDS disease system at organ level as the level of multiscale observation and the within-host scale as the scale of analysis. HIV provirus can be eliminated in specific anatomical compartments/organs of the human body such as the blood by using antiviral drugs while corresponding elimination is not achieved in other anatomical compartments/organs of the human body such as the brain or the central nervous system. As a result, HIV-infected individuals must remain indefinitely on anti-retroviral therapy (or

until a cure is found which can eradicate the virus at within-host scale as the scale of analysis) because of the reciprocal influence between persistence of the virus at within-organ scale and dispersal/spread of virus at between-organ scale. For the organ level, this secondary multiscale loop involves global exchange of pathogen. The organisms exchanged through global exchange in a secondary multiscale loop are either single pathogen species or single pathogen strain. At higher levels of organization of an infectious disease system, this secondary multiscale loop exists when the within-community scale is used as the scale of analysis and the community level is used as the level multiscale observation. In this case there will be reciprocal influence between persistence of infection at microscale (within-community scale) and spread of infection at macroscale (between-community scale). This happens when infection spreads at between-community scale (influenced by persistence of infection at within-community scale) to communities with higher levels of susceptibility or persistence, and being re-introduced back after a period of time at within-community scale (influenced by dispersal/spread of pathogen at between-community scale). In particular, this secondary multiscale loop exists when either (i) the community level I is used as the level of multiscale observation at within-community scale II or within-community scale III or within-whole world scale as the scale of analysis, or (ii) community level II is used as the level of multiscale observation at within-commnity scale III or within-whole world scale as the scale of analysis, or (iii) community level III is used as the level of multiscale observation at within-whole world scale as the scale of analysis. For example, this secondary multiscale loop exists when the within-whole world scale is used as a scale of analysis and the national level is used as the level multiscale observation. In this case the reciprocal influence between persistence of infection at within-nation scale and dispersal/spread of infection at between-nation scale can sustain the infection at within-whole world scale as a scale of analysis. This happens when infection spreads at between-nation scale (influenced by persistence of infection at within-nation scale) to nations with higher levels of susceptibility or persistence, and being re-introduced back after a period of time at within-nation scale (influenced by dispersal/spread of pathogen at between-nation scale). Consider, for example, malaria disease system at national level as the level of multiscale observation with the within-whole world scale as the scale of analysis. Malaria has been eliminated in specific countries of the world by using existing malaria interventions while corresponding elimination has not been achieved in other countries of the world [9]. As a result, countries that have eliminated malaria must remain indefinitely on alert of imported infections (or until malaria is eradicated at within-whole world scale as the scale of analysis) because of the reciprocal influence between persistence of malaria at within-country scale and dispersal/spread of malaria at between-country scale. For the community level this secondary multiscale involves global exchange of pathogen and/or infected hosts and/or their contaminated products. This global exchange of organisms implicated in the transmission of infectious disease systems happens in the context of single pathogen species/strain and single host species. In the context of this study, the two levels of multiscale observation (the organ level, the community level) are called secondary levels of multiscale observation.

c. *Tertiary multiscale loop/cycle*: Depending on the level of organization of an infectious disease system used as the level of multiscale observation in the development of a multiscale model, this multiscale loop in the dynamics of infectious disease systems can either be an infection persistence-dispersal multiscale loop/cycle or a pathogen replication-transmission multiscale loop/cycle. Unlike the other two multiscale loops (the primary

multiscale loop and the secondary multiscale loop) in which the organisms exchanged between levels and scales of an infectious disease system are single pathogen species/ strain and/or single host species infected with single pathogen species/strain, in a tertiary multiscale loop the organisms exchanged between levels and scales of an infectious disease system are multiple pathogen species/strains and/or multiple host species infected with multiple pathogen species/strains and/or multiple host species products contaminated with multiple pathogen species/strains. Therefore, unlike the primary multiscale loop or the secondary multiscale loop which consists of a single loop corresponding to the dynamics of a single pathogen species or a single pathogen strain or a single infected host species, the tertiary multiscale loop is a set of loops corresponding to the dynamics of multiple pathogen species or multiple pathogen strains or multiple infected host species. Further, the tertiary multiscale loop occurs when the microecosystem level (i.e. the microecosystem level I, the microecosystem level II, the microecosystem level III) or the macroecosystem level (i.e. the macroecosystem level I, the macroecosystem level II, the macroecosystem level III) is used as the level of multiscale observation in the development of multiscale models of infectious disease systems. At the two sub-levels of microecosystem level (i.e. the microecosystem level I, the microecosystem level II) this tertiary multiscale loop occurs when the microecosystem level I or the microecosystem level II is used as the level of multiscale observation in the development of multiscale models of infectious disease systems. This multiscale loop involves local exchange pathogen. At these two sub-levels of the microecosystem level, this tertiary multiscale loop involves the reciprocal influence between pathogen replication at the microscale of these levels of organization of an infectious disease system (i.e. within-microecosystem scale I, within-microecosystem scale II), and pathogen transmission at the macroscale of these levels of organization of an infectious disease disease system (i.e. between-microecosystem scale I, and between-microecosystem scale II). At each of these sub-levels of the microecosystem level (i.e. the microecosystem level I, the microecosystem level II), the only organisms exchanged between scales and levels of an infectious disease system are either multiple pathogen species or multiple pathogen strains. At the microecosystem level III this tertiary multiscale loop exists when the within-host scale is used as the scale of analysis and the microecosystem level III is used as the level multiscale observation. In this case there will be reciprocal influence between persistence of infection at microscale (within-microecosystem scale III) and dispersal/spread of infection at macroscale (between-microecosystem scale III). This happens when pathogen spread at between-microecosystem scale III (influenced by persistence of pathogen at within-microecosystem scale III) to microecosystems III with higher levels of susceptibility or persistence, and being re-introduced back after a period of time at within-microecosystem scale III (influenced by dispersal/spread of pathogen at between-microecosystem scale III) and involves global exchange of multiple pathogen species or multiple pathogen strains. Similarly, for the macroecosystem level (macroecosystem level I, macroecosystem level II, macroecosystem level III), this tertiary multiscale loop exists in three main different scenarios as follows. (i) When the macroecosystem level I is used as the level of multiscale observation at within-macroecosystem scale II, or within-macroecosystem scale III, or within-whole world scale as the scale of analysis. (ii) When macroecosystem level II is used as the level of multiscale observation at within-macroecosystem scale III, or within-whole world scale as the scale of analysis. (iii) When macroecosystem level III is used as the level of multiscale observation at within-whole world scale as the scale of analysis. In this case there will be reciprocal influence between persistence of infection at microscale (within-macroecosystem scale I, within-macroecosystem scale II, within-

macroecosystem scale III) and dispersal/spread of infection at macroscale (between-macroecosystem scale I, between-macroecosystem scale II, between-macroecosystem scale III). The tertiary multiscale loop at the macroecosystem level involves global exchange of either (i) multiple pathogen species or (ii) multiple pathogen strains or (iii) multiple infected host species or (iv) their products. In the context of this study, the two levels of of multiscale observation (the microecosystem level, the macroecosystem level) are called tertiary levels of multiscale observation.

These three multiscale loops (primary mltiscale loop, secondary multiscale loop, tertiary multiscale loop) are the basis for a broader scientific theory for multiscale modelling of infectious disease systems established in [2] which states that at every level of organization of an infectious disease system there is no privileged/absolute scale which would determine disease dynamics, only interactions between the microscale and macroscale. This is because at every level of organization of an infectious disease system, its multiscale dynamics is governed by at least one of these multiscale loops which involve the reciprocal influence between the microscale and the macroscale. We shall discuss further the usefulness of multiscale models with these different multiscale loops/cycles in this article in relation to control, elimination, and eradication of infectious disease systems.

### The local and global exchange of organisms implicated in disease dynamics result in five main different linking mechanisms of levels and scales of infectious disease systems

The local and global exchange of organisms implicated in the dynamics of an infectious disease system results in biological linkage of levels and scales of an infectious disease system. Based on these local and global exchange of organisms implicated in the dynamics of an infectious disease system, we identify five main biological linking mechanisms of the levels and scales of an infectious disease system which are: (a) linked through exchange of single pathogen species/strain, (b) linked through exchange of multiple pathogen species/strains, (c) linked through exchange of single infected host species/host species products, (d) linked through exchange of infected multiple host species/host species products, and (e) linked through exchange of mobile genetic elements such as phage and plasmids.

Fig 2 shows four of the five main biological linking mechanisms of the levels and scales of an infectious disease system. In Fig 2, each of the four biological linking mechanisms is represented by a circular shape drawn with dotted lines with one side of the circular shape cutting through the levels of organization of an infectious system linked by that mechanism. The linkage of levels and scales of an infectious disease system through the fifth linking mechanism, i.e through exchange of mobile genetic elements such as phage and plasmids is not shown in Fig 2 in order to keep it simple and easier to understand. In what follows we briefly describe each of the five biological linking mechanisms of the levels and scales of an infectious disease system.

 a. *Linked through exchange of single pathogen species/strain*: As shown in Fig 2, this linking mechanism of the levels and scales of an infectious disease system through exchange of the single pathogen species/strain is associated with five levels of organization of an infectious disease system (the cell level, the tissue level, the organ level, the host level, and the community level). However, at three of the lower levels of organization of an infectious disease system (the cell level, the tissue level, the host level), this biological linking mechanism of levels and scales of an infectious disease system happens through local exchange of pathogen and involves a primary multiscale loop/cycle. But at two

other levels of organization of an infectious disease system which are the organ level and the community level (i.e. community level I, community level II, and community level III), this linkage of levels and scales of an infectious disease system happens through global exchange of single pathogen species/strain and involves a secondary multiscale loop.

b. *Linked through exchange of multiple pathogen species/strains*: As illustrated in Fig 2, this linkage of levels and scales of an infectious disease system is associated with two levels of organization of an infectious disease system which are the microecosystem level with three sub-levels (i.e. microecosystem level I, microecosystem level II, and microecosystem level III), and the macroecosystem level with three sub-levels also (i.e. macroecosystem level I, macroecosystem level II, and macroecosystem level III). At the microecosystem level and macroecosystem level, this linking mechanism involves a tertiary multiscale loop. At two of the sub-levels of the microecosystem which are microecosystem level I and microecosystem level II this biological linking mechanism of levels and scales of an infectious disease system happens through local exchange of multiple pathogen species/strains. At the microecosystem level III, this linking mechanism of levels and scales of an infectious disease system involves global exchange of multiple pathogen species/strains. Similarly, at the sub-levels of the macroecosystem level which are (i) the macroecosystem level I (which is the local community level considered as an ecosystem), (ii) the macroecosystem level II (which is the territorial community level considered as an ecosystem), and (iii) the macroecosystem level III (which is the regional community level considered as an ecosystem), this biological linking mechanism of levels and scales of an infectious disease system happens through global exchange of multiple pathogen species/strains.

c. *Linked through exchange of host species/host species products infected/contaminated with single pathogen species/strain*: This linking mechanism of the levels and scales of an infectious disease system through exchange of the single infected host species and/or their products (meat, fruits, vegetables, etc.) occurs only through global exchange of infected host or their products and involves a secondary multiscale loop. Fig 2 shows that this biological linking mechanism is associated with community level (i.e. community level I, community level II, and community level III). For this level of organization of an infectious disease system, the infected hosts or their products are infected with single pathogen species/strain.

d. *Linked through exchange of multiple host species/host species products infected/contaminated with multiple pathogen species/strains*: From Fig 2, we notice that this biological linking mechanism is associated with the macroecosystem level (i.e. macroecosystem level I, macroecosystem level II, and macroecosystem level III). This mechanism of linking levels and scales of an infectious disease system which happens through global exchange of host species and/or their products infected/contaminated with multiple pathogen species/strains also involves a tertiary multiscale loop.

e. *Linked through exchange of mobile genetic elements*: This happens through processes of horizontal gene transfer, e.g., via phage or plasmids, which can occur between pathogen species that occupy the same niche/host and between strains of the same species during co-infection [4, 6]. This linking mechanism can be considered at various levels of organization of an infectious disease from the lowest level of the infectious disease system which in the context of bacterial infections can be from the cell level (i.e. within-bacterial cell and between-bacterial cell) where local exchange of mobile genetic elements such as

phage and plasmids takes place to the highest level of organization of an infectious disease system which is the macroecosystem level where global exchange of mobile genetic elements takes place. This linking mechanism of the levels and scales of an infectious disease system is not illustrated in Fig 2 in order to keep it simpler.

Therefore, in general, scales associated with lower levels of organization of an infectious disease system (cell level, tissue level, organ level, microecosystem level, and host level) are linked through exchange of pathogen only while scales associated with higher levels of organization of an infectious disease system (community level and macroecosystem level) are linked through exchange of infected host/host products as well as exchange of pathogen. The key challenge in the development of multiscale models at each of the seven main levels of organization of an infectious disease system is to find appropriate quantitative (mathematical, statistical, computational, algorithmic, etc.) ways of representing these biological mechanisms of linkage between the scales of an infectious disease system.

## Results

The proposed research and development process for multiscale models of infectious disease systems informed by the multiscale vision of infectious disease systems which we have just presented in section involves an iterative scheme between four stages. Fig 3 shows a schematic diagram of the four-stage research and development process for multiscale models of infectious disease systems. The four-stage process model presented schematically in Fig 3 gives consistency in the way the research and development process for multiscale models of infectious disease systems is carried out by scientists with different skills (e.g. biology, epidemiology, medicine, microbiology, mathematical modelling, statistical modelling, pharmacology, ecology, etc.). The essential four stages for the process of development of multiscale models of infectious disease systems are further described in what follows.

### Stage I: Define the infectious disease problem to be addressed by the multiscale modelling study

This stage consists of four steps which we describe as follows.

1.1. **State the objective of multiscale modelling**: The first step in this first stage involves defining the objective of the multiscale modelling task. This step gives a clear statement of the decision to be supported by the multiscale model. It specifies details of the information required from the multiscale model. The initial specification of the objective of the multiscale modelling of the infectious disease system with regard to detail may later on be modified on the basis of information obtained from the other three subsequent stages (stage II, State III, and stage IV).

1.2. **Decide on levels of organization to include into the multiscale model and their associated scales**: The consideration of levels of organization of an infectious disease system to be incorporated into the multiscale model help us to simplify the perceived complexity of infectious disease systems. This also helps to identify the type of multiscale model to be developed which can be any of the following seven types: cL-MSMs (for the cell level), TL-MSMs (for the tissue level), OL-MSMs (for the organ level), mL-MSMs (for the microecosystem level), HL-MSMs (for the host level), CL-MSMs (for the community level), ML-MSMs (for the macroecosystem level). The level(s) of organization of an infectious disease system to be incorporated into the multiscale model are determined by the research question to be addressed. Failure to specify the level(s) of organization of

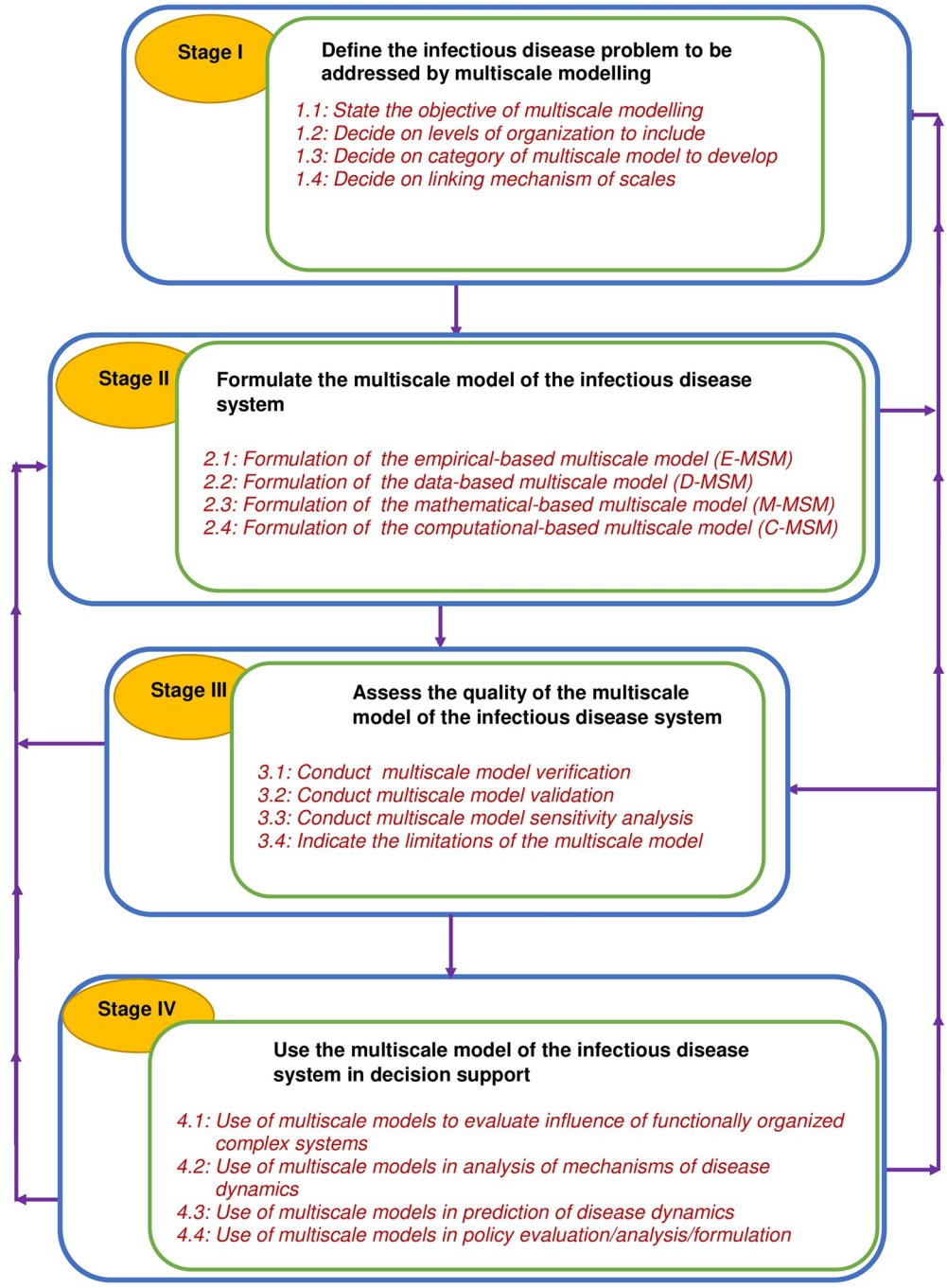

**Fig 3. The schematic diagram of the four-stage research and development process for multiscale models of infectious disease systems which are as follows.** *Stage I*: define the infectious disease problem to be addressed by multiscale modeling, *Stage II*: formulate the multiscale model of the infectious disease system, *Stage III*: assess the quality of the multiscale model of the infectious disease system, and *Stage IV*: use the multiscale model of the infectious disease system in decision support with four steps for each of the four stages. The arrows in this Figure indicate the iterative process of moving from one development stage of the multiscale model to another.

an infectious disease system incorporated into the multiscale model is a critical deficiency in some multiscale modelling studies of infectious disease systems. In deciding on level(s) of organization of an infectious disease system to be incorporated into the multiscale model, it is important to make distinction between the different types of the levels of organization of an infectious disease system which are: (i) levels of multiscale analysis, and (ii) levels of multiscale observation. The most crucial aspect in this step is to select the level of multiscale observation which is usually any of the seven levels of organization of an infectious disease system ((i) the cell level, (ii) the tissue level, (iii) the organ level, (iv) the microecosystem level, (v) the host level, (vi) the the community level, and (vii) the macroecosystem level) and the scale of analysis which is usually any of the following seven: (i) within-tissue scale, (ii) within-organ scale, (iii) within-microecosystem scale (i.e. within-microecosystem scale I, within-microecosystem scale II, within-microecosystem scale III), (iv) within-host scale, (v) within-community (i.e. within-community scale I, within-community scale II, and within-community scale III), (vi) within-macroecosystem scale (i.e. within-macroecosystem scale I, within-macroecosystem scale II, and within-macroecosystem scale III), and (vii) within-whole world scale. Once the level of multiscale observation and the scale of analysis to be used in the development of a multiscale model are decided on with clear specification of the boundaries of the scales of observation, then decisions on the details of the category of multiscale model to be developed follow.

1.3. **Decide on category of multiscale model to develop**: To make multiscale modelling of infectious disease systems more applicable/useful, the suitability and feasibility of different categories of multiscale models need to be understood. The choice of the category of the multiscale model to be developed is often important as it, to some extent, determines: (i) the level of mechanistic detail and knowledge to be incorporated into the multiscale model, (ii) the easiness of constructing the multiscale model, (iii) the level of heterogeneity that can be incorporated into the multiscale model, (iv) the easiness for researchers to know where to start when first creating their own multiscale model, and (v) the easiness to describe the multiscale model by referring to the generic description of the category. There are five main generic categories of multiscale models of infectious disease systems to which each of the seven different types of multiscale models (cL-MSMs, TL-MSMs, OL-MSMs, mL-MSMs, HL-MSMs, CL-MSMs, ML-MSMs) may belong to, which are [1, 2, 5]: (a) individual based multiscale models (IMSMs), (b) nested multiscale models (NMSMs), (c) embedded multiscale models (EMSMs), (d) hybrid multi-scale models (HMSMs), and (e) coupled multiscale models (CMSMs). While it is difficult at this stage to draw general conclusions about the suitability of each of the five categories of multiscale models of infectious disease systems (IMSMs, NMSMs, EMSMs, HMSMs and CMSMs), from the kind of empirical studies from which they were determined [1, 5], some general implications of selection of a particular category of multiscale models are as follows:

a. *For IMSMs*: Selection is dictated by the need to incorporate heterogeneity (e.g. (i) heterogeneity in host susceptibility to infection, (ii) heterogeneity in the ability of hosts to transmit pathogens to other hosts, (iii) heterogeneity in host immune response, (iv) heterogeneity in host behaviour) into the multiscale model. But the incorporation of heterogeneity into the multiscale model comes at the cost of increased computational burden in solving the multiscale model. The major weakness of IMSMs is that although they can represent the microscale explicitly, they do not explicitly represent the macroscale for each level of organization of an infectious disease used as the level of multiscale

observation in the development of a multiscale model [10]. The microscale submodel results are usually converted by summing up, averaging, or performing some statistical analysis of them into macroscale variables for interpretation at that scale [5]. However, IMSMs have an advantage over the other categories of multiscale models (NMSMs, EMSMs, HMSM, CMSMs) in that they require little mathematical expertise. The majority of IMSMs are developed using agent based modelling methods. For details on development of IMSMs using agent-based techniques see [10] and references therein. However, other IMSMs are developed using graph-theoretic methods. Typical examples of IMSMs developed using graph-theoretic methods are [11, 12].

b. *For NMSMs*: Selection is dictated by the choice of biological linking mechanism between the microscale and the macroscale in which the influence of the macroscale on the microscale through super-infection is considered negligible/insignificant and its effects on disease dynamics in the multiscale model can be ignored. Nested multiscale models are normally selected for modelling infectious disease systems in which the pathogen has a replication cycle at the microscale (e.g. within-host scale). The multiscale model for human HIV/AIDS and influenza A virus in [13, 14] at host level, and in [15] at cell level are good examples of nested multiscale models. Another specific case where nested multiscale models are more appropriate is for infectious disease systems in which the pathogen replicates at both the microscale (e.g. within-host scale) and at the macroscale (between-host scale). Such infectious disease systems are caused by opportunistic infections such as cholera, Salmonella enterica and anthrax. For these infectious disease systems a nested multiscale model is more appropriate than an embedded multiscale because the contribution of super-infection to pathogen load at the microscale is negligible compared to the contribution of the pathogen replication cycle. The multiscale model for cholera in [16] at host level is a good example of such nested multiscale models. Nested multiscale models facilitate easy of reduction of the dimensions of the multiscale model, for example through slow and fast time scale analysis [13], making it easier to analyze the multiscale model.

c. *For EMSMs*: Selection is dictated by the choice of biological linking mechanism between the microscale and the macroscale in which the influence of the macroscale on the microscale through super-infection is considered important/significant and its effects on disease dynamics in the multiscale model can no longer be ignored. For such infectious disease systems, the pathogen does not have a replication cycle at microscale. Infectious disease systems where the pathogen (infectious agent) does not have a replication cycle at microscale are modelled by embedded multiscale models. Examples of such infectious diseases are soil transmitted helminths infections such as hookworm. This is particularly true for helminths infections because, with few exceptions (Strongyloides, Trichinella, tapeworm larvae), helminths do not have a replication cycle at the microscale (e.g within-host scale). For these infectious disease systems, the pathogen load at the within-host scale increases through super-infection (i.e. repeated infection before the host recovers from an infectious episode). Embedded multiscale models do not easily get reduced in dimension making it more difficult to analyze them compared to nested multiscale models. The multiscale modeling of hookworm in [2] illustrates a typical embedded multiscale model in which the influence of the macroscale on the microscale is through super-infection.

d. *For HMSMs*: Selection is dictated by the freedom to represent the microscale and the macroscale using different mathematical representations for each level of organization

of an infectious disease used as the level of multiscale observation in the development of a multiscale model. Therefore, HMSMs take the form of either NMSMs or EMSMs except that in this case the microscale and the macroscale have different mathematical formulations. Typical examples of such paired formalisms are (i) deterministic/stochastic, (ii) discrete time/continuous time, (iii) mechanistic/phenomenological, (iv) ODE/PDE, etc. Currently, the majority of HMSMs are developed using the ODE/PDE formalism and are based on the methodology established in [17]. The multiscale model for paratuberculosis in [18] developed using the host level as the level of multiscale observation, and for HIV in [19] developed using the cell level as the level of multiscale observation are good examples of hybrid multiscale models developed using the method in [17]. However, the freedom to represent the different scales (microscale and macroscale) using different mathematical representations comes at a cost associated with the difficulty to analyze the multiscale model because the different scales would require different solution methods (analytical or numerical solutions).

e. *For CMSMs*: Selection is dictated by the need to incorporate multiple levels of organization of the infectious disease system, and/or multiple host species such as in vector-borne diseases, and/or multiple pathogen species/strains such as happens in co-infection, and/or multiple communities, and/or multiple anatomical compartments/organs in the development of multiscale models of infectious disease systems. As a result, coupled multiscale models are developed using the other four categories of multiscales (IMSMs, NMSMs, EMSMs, HMSMs) as sub-models. The multiscale model for human schistosomiasis in [20], Guinea worm in [21], human onchocerciasis in [22], and malaria [23, 24] are typical examples of coupled multiscale models. These coupled multiscale models were developed using either (i) a combination of a nested multiscale for the host where pathogen has a replication cycle at microscale (i.e. within-host scale) and an embedded multiscale for the host where pathogen does not have a replication cycle at microscale (i.e. within-host scale) as in malaria [23] (where pathogen does not have a replication cycle in the mosquito vector but has a replication cycle in the human host) or (ii) a combination of embedded multiscale models where pathogen does not have replication cycle at microscale in both hosts (human host and vector host) as in human onchocerciasis [22]. But other coupled multiscale models are also been developed using HMSMs as sub-models (see [25] and references therein).

Therefore, the most appropriate category of multiscale model for a given multiscale modelling problem of an infectious disease system is determined by a number of factors, some of which are still to be established. In future, research needs to be conducted to establish a more elaborate guiding framework for selecting a suitable candidate category of a multiscale model.

1.4. **Decide on linking mechanism of scales of an infectious disease system**: This step should start by establishing which of the five main biological linking mechanisms ((i) exchange of single pathogen species/strain, (ii) exchange of multiple pathogen species/strains, (iii) exchange of single infected host species/host species products, (iv) exchange of infected multiple host species/host species products, (v) exchange of mobile genetic elements such phage and plasmids) is relevant for developing a multiscale model at a particular level of multiscale observation of an infectious disease system. While the guidance offered by the categorization of multiscale models at all the seven main levels of organization of an infectious disease system [1, 5] may help the modeler conceptually in understanding the nature of biological linking mechanisms between the scales, there are still challenges with the implementation of the conceptual ideas at the practical level in

developing multiscale models of infectious disease systems. Once a decision is made on the nature of the biological linking mechanism, then the next task is to establish the method of representing the biological linking mechanisms in the development of a particular category of a multiscale model (IMSM, NMSM, EMSM, HMSM, CMSM). There is no unique way of representing the different biological linking mechanisms of the levels and scales of an infectious disease system ((i) exchange of single pathogen species/strain, (ii) exchange of multiple pathogen species/strains, (iii) exchange of single infected host species/host species products, (iv) exchange of infected multiple host species/host species products, (v) exchange of mobile genetic elements such phage and plasmids). Efforts to represent these biological linking mechanisms in the development of multiscale models that belong to the different categories (IMSM, NMSM, EMSM, HMSM, CMSM), have yielded a number of different multiscale models such as (i) [10–12] for IMSMs, which are linked through exchange of single pathogen species/strain based on microscale inspired macroscale models approach [10] and graph-theoretic methods [11, 12], (ii) [20–25] for CMSMs in which the microscale and the macroscale are linked through exchange of single pathogen species/strain, (iii) [13, 26], for NMSMs in which the microscale and the macroscale are also linked through exchange of single pathogen species/strain, and (iv) [19, 25] for HMSMs in which the microscale and the macroscale are linked through time-since-infection approach as well as through exchange of single pathogen species/strain. More research is needed to establish how to represent the different biological forms of linkage between scales in the development of multiscale models of infectious disease systems, particularly when taking into account both local and global exchange of organisms implicated in infectious disease dynamics together with the different multiscale loops/cycles (primary multiscale loop, secondary multiscale loop, tertiary multiscale loop) involved in each category of multiscale models.

## Stage II: Formulate the multiscale model of the infectious disease system

This stage builds on the knowledge gathered in stage I about the multiscale model to be developed and involves the actual development of the multiscale model. The proposed actual development of the multiscale models of infectious disease systems involves integration of knowledge from four different multiscale modelling approaches. Multiscale modelling of infectious disease systems has been defined in [5] as any representation of an infectious disease system that can be used to study or characterize an infectious disease system at more than one scale. We can identify four different multiscale modelling approaches which satisfy the definition in [5] which are: (i) empirical-based multiscale models (E-MSMs), (ii) mathematical-based multiscale models (M-MSMs), (iii) computational-based multiscale models (C-MSMs), and (iv) data-based multiscale models (D-MSMs). Fig 4 shows a conceptual diagram of the integration of knowledge from the four different multiscale modelling approaches. Although the formulation of the different multiscale models (E-MSMs, M-MSMs, C-MSMs, D-MSMs) require interdisciplinary insights, we briefly describe the formulation of each multiscale modelling approach separately as follows.

2.1. **Formulation of empirical-based multiscale models (E-MSMs)**: These are multiscale models of infectious disease systems based on experimental systems, culture systems, clinical trial systems, observational systems or surveillance systems that characterize an infectious disease system at more than one scale. Empirical multiscale modelling involves data collection from the various scales of observation used in the development of the multiscale model. This requires development of methods, practices, and devices

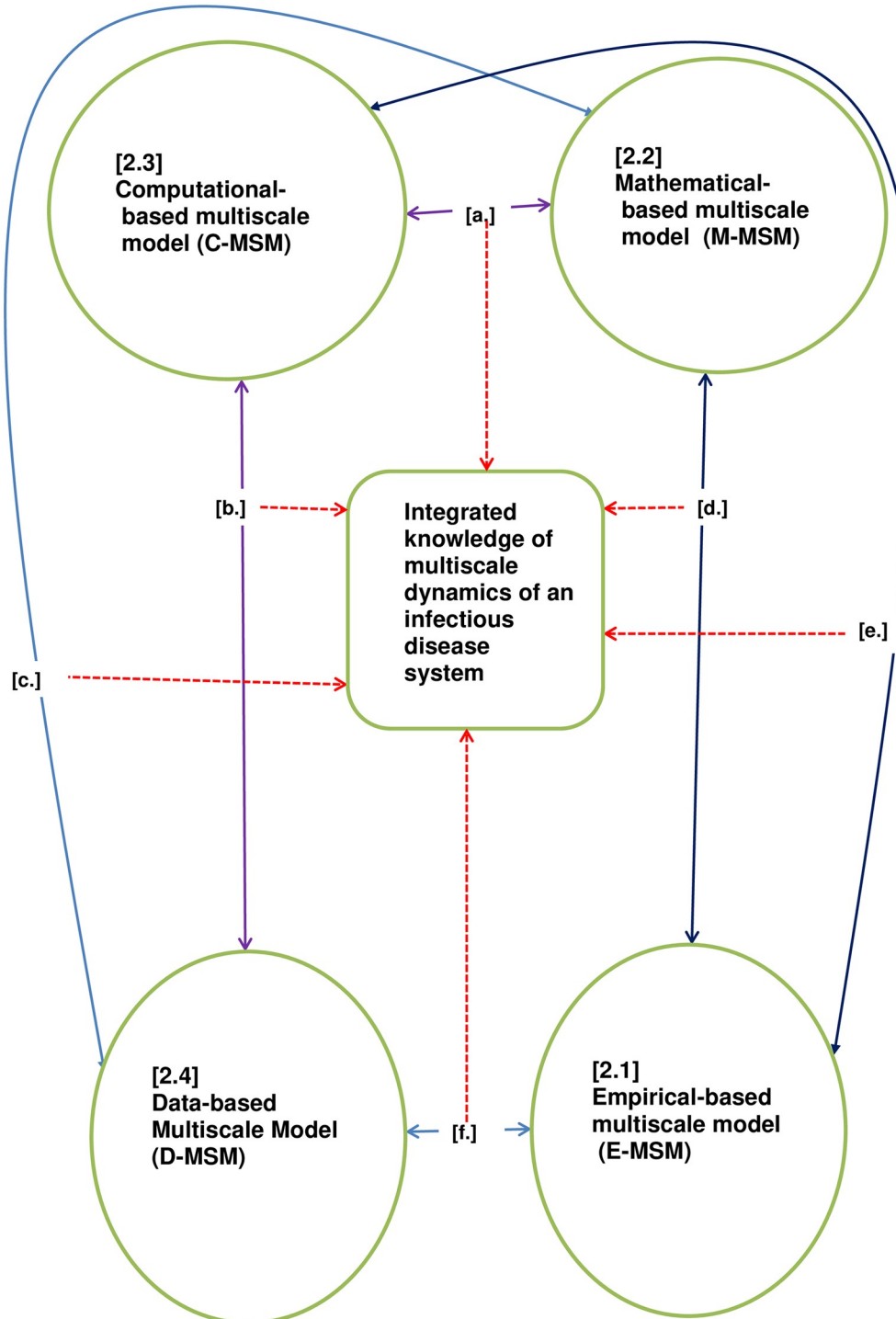

**Fig 4. A conceptual diagram of the integration of knowledge from the four different multiscale modelling approaches of infectious disease systems: [2.1] empirical-based multiscale model (E-MSM), [2.2] mathematical-based multiscale model (M-MSM), [2.3] computational-based multiscale model (C-MSM), and [2.4] data-based multiscale model (D-MSM) and the interplay between the four different multiscale modelling approaches when considered in pairs.** [a.] interplay between M-MSMs and C-MSMs, [b.] interplay between D-MSMs and C-MSMs, [c.] interplay between M-MSMs and D-MSMs, [d.] interplay between M-MSMs and E-MSMs, [e.] interplay between C-MSMs and E-MSMs, and [f.] interplay between D-MSMs and E-MSMs.

for measuring, observing, defining and characterizing infectious disease systems at specific scales. Empirical multiscale modelling is foundational to all the other three multiscale modelling approaches (M-MSMs, C-MSMs, D-MSMs). The difficulty of developing E-MSMs arise from both technical considerations as well as ethical and regulatory considerations, especially for animal-based and human-based E-MSMs.

2.2. **Formulation of mathematical-based multiscale models (M-MSMs)**: This involves the use of mathematical and statistical approaches to develop multiscale dynamical models. For the M-MSMs, this approach begins with listing all the equations of the different sub-models describing each scale, a description of the meaning of each equation, variable, and parameter in each equation. A possible summary list of all parameters and variables associated with each scale incorporated into the multiscale model. It also involves establishing casual links between the variables as well as the casual links between the scales.

2.3. **Formulation of computational-based multiscale models (C-MSMs)**: This involves developing computational algorithms to translate M-MSMs and D-MSMs into C-MSMs. Some individual-based multiscale models are also C-MSMs but are not derived from M-MSMs or D-MSMs. A computer program used to derive approximate/ numerical solutions to D-MSM or M-MSM is itself a C-MSM. This multiscale modelling approach focuses on development of generic methods, algorithms, specification and software environments for multiscale computing applicable in any scenario where M-MSMs and D-MSMs require non-trivial computing solutions.

2.4. **Formulation of data-based multiscale models (D-MSMs)**: The power of D-MSMs lie within leveraging their strength for transforming multiscale data into knowledge about the dynamics of infectious disease systems. This multiscale modelling approach involves working with, interpreting, visualizing or making sense of multiscale data (empirical multiscale data from E-MSMs or predicted multiscale data from C-MSMs) generated at different levels and scales of an infectious disease system. Therefore, data-based multiscale modelling involves development of methods and technologies for organizing, managing, curating, processing, integrating, analyzing e.g. through machine learning, data warehousing, data and information modelling, information retrieval, mining and sharing of multiscale data.

The four different multiscale modelling approaches can further be demarcated into two main groups: (i) the E-MSMs, which stand out as a separate group because they cannot be used as predictive tools, and (ii) the quantitative multiscale models (Q-MSMs), which include the other three multiscale modelling approaches (M-MSMs, C-MSMs, and D-MSMs) which can be used as predictive tools. To further bring some order in the discussion of multiscale models of infectious disease systems, we propose that the categorization of multiscale models of infectious disease systems [1, 5] originally developed for M-MSMs only be applicable to all the other three multiscale modelling approaches (i.e. E-MSMs, C-MSMs, and D-MSMs).

The integration of knowledge from the four different multiscale modelling approaches is perhaps best illustrated by a diagram, which portrays the interdisciplinary nature of the interplay between the different multiscale modelling approaches by considering them in pairs and then threesomes in order to integrate knowledge from the different multiscale modelling approaches. Fig 4 is a conceptual representation of the interplay between the four different multiscale modelling approaches of infectious disease systems. The interplay between the four different multiscale modelling approaches is now briefly discussed when considered in pairs.

a. *Interplay between M-MSMs and C-MSMs*: In general, a M-MSM alone is not enough to characterize the dynamics of an infectious disease system because it will not be possible to derive closed form solutions of the M-MSM. In most cases, our best efforts to obtain solutions of the M-MSM will only yield approximate solutions which are obtained from C-MSMs. Therefore, in this case, M-MSM is build first and is then used to build a C-MSM which is then used to study the M-MSM by numerical simulation.

b. *Interplay between D-MSMs and C-MSMs*: Similarly, a D-MSM alone is not enough to characterize the dynamics of an infectious disease system and will require translation of the D-MSM using computational algorithms into a corresponding C-MSM which is then used to simulate the data.

c. *Interplay between M-MSMs and D-MSMs*: M-MSMs determine how the system changes from one state to the next and describes the interdependence of the variables and scales involved. D-MSMs characterize the numerical multiscale data and may be used to estimate probabilistic future behaviour of an infectious disease system based on its past behaviour characterized by the multiscale data. Although M-MSMs and D-MSMs are usually developed to address different questions under different assumptions in the multiscale dynamics of infectious disease systems, their synergy gives an added value to the overall multiscale description of an infectious disease system because the exchange of information between D-MSM and M-MSM involves identifying knowledge that can be extracted from each multiscale modelling approach. However, D-MSMs are more flexible than M-MSMs as they can be changed as per arrival of new multiscale data and therefore can easily incorporate new and emerging patterns and trends through techniques such as pattern recognition/machine learning.

d. *Interplay between M-MSMs and E-MSMs*: In multiscale modelling studies, multiscale data from E-MSM is needed to parametrize the M-MSM. Further, empirical data from the E-MSM is often used in the validation of the M-MSM (see stage III in this research and development process for multiscales of infectious disease systems).

e. *Interplay between C-MSMs and E-MSMs*: The interplay between C-MSMs and E-MSMs is usually more or less the same as the interplay between M-MSMs and E-MSMs. This is because, in most cases, a M-MSM is usually converted into a C-MSM with the same parametrization as the M-MSM with multiscale data from E-MSM. Further, in the validation of the M-MSM, agreement between predicted multiscale data from C-MSM with empirical multiscale data from E-MSM is often taken to imply that the M-MSM is valid and may be used in decision support.

f. *Interplay between D-MSMs and E-MSMs*: Data from E-MSM is often used to construct the D-MSM.

The interplay between the four different multiscale modelling approaches enables the development of multiscale models of infectious disease systems based on interdisciplinary insights. However, the interplay between M-MSMs and E-MSMs has recently been investigated [27]. But, this was only at one level of organization of an infectious disease system, i.e. the host level. More research is need to identify the full extend of the interplay between the different multiscale modeling approaches for the different levels of organization of infectious diseases and how this can be optimized to enhance interdisciplinary efforts in the development of multiscale models of infectious disease dynamics.

## Stage III: Evaluate the quality of the multiscale model of the infectious disease system

This stage can result in a cycle of refinements of stage I, stage II, stage III, and stage IV until a sound version of the multiscale model is reached at this stage (stage III). One of the key requirements in research and development process for multiscale models of infectious disease systems at this stage (stage III) is to check and reduce sources of errors in the multiscale model. There are two main sources of errors.

a. *Sources of errors from individual sub-models that describe each scale of an infectious disease system*: These sources could be from (i) the single scale sub-model structure (e.g. due to the nature of the casual links between the variables), and/or (ii) the single scale sub-model parameters, and/or (iii) the single scale sub-model computational/numerical algorithm or solution method.

b. *Sources of errors from full multiscale model that describes the integrated scales of an infectious disease system*: These sources of error could be due to (i) down-scaling and up-scaling of parameters and variables when linking/integrating the single scale sub-models into a complete multiscale model (i.e. errors from the nature of casual links between the scales), and/or (ii) simplifying the multiscale model such as reduction of the dimensions (order) of the full multiscale model to try to improve computational efficiency, and/or (iii) heterogeneity of the sub-models that describe the different scales.

In order to detect and reduce these errors, the following four steps should be undertaken at this stage:

3.1. **Conduct multiscale model verification**: This step consists of checking that each of the multiscale modelling approaches (M-MSM, D-MSM, C-MSM, E-MSM) is doing what it is intended to do. It involves the process of checking if each of the multiscale modelling approaches (M-MSM, D-MSM, C-MSM, E-MSM) produces results that are expected from it. The verification would be different for different multiscale modelling approaches. For example, for M-MSM, this would involve checking if it is mathematically and biologically well-posed, and that the casual relationship between the variables and between the scales are correctly represented. But, in general, this step involves checking and ensuring that the four multiscale modelling approaches (M-MSM, D-MSM, C-MSM, E-MSM) incorporate known biological and epidemiological knowledge and theories. Here the multiscale modelling approaches (M-MSM, D-MSM, C-MSM, and E-MSM) should be examined to ensure that all the main known determinants that influence the infectious disease system are included into the multiscale model.

3.2. **Conduct multiscale model validation**: This step consists of checking if the different multiscale modelling approaches (M-MSM, D-MSM, C-MSM, and E-MSM) are in agreement. Therefore, this step involves establishing that the Q-MSMs (M-MSM, D-MSM, C-MSM) behave sufficiently similar to E-MSM. This can be achieved by checking and ensuring that the predicted multiscale data from Q-MSMs (M-MSM, D-MSM, C-MSM) behaves sufficiently similar to empirical data from E-MSM. Therefore, this step involves checking and ensuring that the output from each of the four multiscale modelling approaches (M-MSM, D-MSM, C-MSM, E-MSM) behaves according to known biological and epidemiological knowledge and theories. However, the main challenge with validating Q-MSMs against empirical data is that they are sometimes constructed to simulate scenarios with which there are no data from E-MSMs or where it is

not feasible to formulate E-MSMs to compare with for many reasons that may include costs or ethical considerations or technical feasibility.

3.3. **Conduct multiscale model sensitivity analysis and uncertainty analysis**: A multiscale model which is highly sensitive to parameters on which there is little reliable multiscale data is of limited use in decision-making. Those parameters to which the Q-MSMs (M-MSM, D-MSM, C-MSM) are sensitive should be the focus of multiscale data collection efforts in E-MSM if the validity and utility of the Q-MSMs (M-MSM, D-MSM, C-MSM) are to be improved. On the other hand, if varying a parameter has little influence on the key outputs of the Q-MSM, uncertainty about the value of that parameter does not detract from the value of the multiscale model in decision support. Finding parameters which are known to vary in an E-MSM and to which the Q-MSMs (M-MSM, D-MSM, C-MSM) are sensitive is also useful in a different way in that they provide guidelines for control, elimination, and eradication of the infectious disease system. If there are parameters which could be influenced by disease control or elimination or eradication activities, then they become 'critical points' which should be monitored and controlled during an infectious disease system outbreak. However, if sensitivity analysis reveals that a Q-MSM (M-MSM, D-MSM, C-MSM) output is more sensitive to particular parameters in the E-MSM, this may suggest that the Q-MSM (M-MSM, D-MSM, C-MSM) is of limited use in decision making. Therefore, in general, there are two different, but related, types of analyses that can be conducted in this third step for a multiscale model of an infectious disease: (i) sensitivity analysis to determine which parameters are the outputs most sensitive to i.e., using partial rank correlation coefficient [28], and (ii) uncertainty analysis to determine how uncertainty in inputs affect uncertainty in outputs i.e., using Latin Hypercube sampling of parameter space [29].

3.4. **Indicate the limitations of the multiscale model**: Like all other modelling approaches, not all multiscale modelling is without limitations. Multiscale models differ substantially in their comprehensiveness, quality and usefulness. The limitations of the multiscale model should be documented.

## Stage IV: Use of multiscale models of infectious disease system in decision support

In general, the research and development process for multiscale models of infectious disease systems scientific agenda has four main items which are all related to the decision to be supported. These include (i) use of multiscale models as a framework for understanding the influence of functionally organized complex systems on infectious disease dynamics, (ii) use of multiscale models as strategic tools in analyzing/understanding the underlying mechanisms of infectious disease dynamics, (iii) use of multiscale models in predicting/forecasting of infectious disease dynamics, and (iv) use of multiscale models in evaluation/analysis/formulation of policy for control or elimination or eradication of an infectious disease system. Specific details of these uses of multiscale models of infectious disease systems are now further explained.

4.1. **Use of multiscale models in the analysis of the influence of some functionally organized complex systems on infectious disease dynamics**: The interaction of the three sub-systems of an infectious disease system (the pathogen sub-system, the environmental sub-system, and the host sub-system) to establish an infectious disease system (a structurally organized complex system) is embedded within a much larger network of six main functionally organized complex systems that influence the multiscale dynamics

of infectious disease systems. A major deficit in our knowledge of the multiscale dynamics of infectious disease systems is the impact of functionally organized complex systems on infectious disease dynamics. Functionally organized complex systems are dominated by multiple mechanisms that influence infectious disease dynamics at different scales. The six main functionally organized complex systems which influence the multiscale dynamics of infectious disease systems are:

a. *Economic system*: A type of functionally organized complex system that consists of two main levels/scales: (i) the microeconomic level/scale which plays out at individual level, the family level, the local community level and (ii) the macroeconomic level/scale which plays out at the national/country level, and the regional level.

b. *Evolutionary system*: A type of functionally organized complex system that consists of two levels/scales: the microevolutionary level/scale and the macroevolutionary level/scale. In the context of this study, microevolutionary level/scale processes take place at the microecosystem level of organization of an infectious disease system (see Fig 2) while the macroevolutionary level/scale processes take place at the macroecosystem level of organization of an infectious disease system (see Fig 2). However, unlike other complex systems in which each level of the complex system is associated with process with different mechanisms, processes at the microecosystem level of organization of an infectious disease system and processes at the macroecosystem level of organization of an infectious disease system are governed by the same four main mechanisms: (i) mutation, (ii) migration/dispersal of pathogen species/strains and/or host species (for the macroecosystem level of an infectious disease system) or infection or super-infection (i.e. repeated infection by pathogen of the same species/strain before the host recovers from prior infection by the same species/strain) together with shedding/excretion (for the pathogen migration/dispersal at the microecosystem level of an infectious disease system), (iii) genetic drift and (iv) natural selection.

c. *Social system*: A type of functionally organized complex system that consists of two main levels/scales: (i) the microsocial level/scale which plays out at the individual human level and (ii) the macrosocial level/scale which plays out at the human society level. This functionally organized complex system includes the cultural, religious, social and behavioural dimensions of infectious disease dynamics.

d. *Environmental change system*: A type of functionally organized complex system that consists of two main levels/scales: the microenvironmental change level/scale, which plays out at the inside-host environmental scale and the macroenvironmental change level/scale, which also plays out at the outside-host environmental scale. The main drivers of microenvironmental change level/scale are drugs and immune response. At the macroenvironmental change level/scale the drivers are either: (i) due to naturally-induced mechanisms or (ii) due to human induced mechanisms. Changes at the macroenvironmental scale such as ambient temperature, rainfall, humidity, land cover, land use, population growth, extreme weather events, natural disasters, climate change or soil moisture can significantly influence the presence and population growth, survival and reproductive capacity of vectors and pathogens at the macroenvironmental scale.

e. *Health interventions system*: A type of functionally organized complex system that consists of two main levels/scales: the medical interventions level/scale and the public health

interventions level/scale. The medical interventions level/scale consists of those health interventions that are curative or healing in mechanism for an existing infection while the public health interventions level/scale are those that are preventive in mechanism for an impending infection. Public health interventions level/scale include a wide range of activities: policies, laws, and regulations; organizational or community developments; education of individuals and communities; engineering and technical developments; service development and delivery; and communication. They can be classified into three broad categories which are: (i) clinical prevention interventions—which are offered by health-care workers and usually in clinical settings and involve using some drugs such as mass drug administration or vaccines, (ii) Behavioural change strategies—which include health promotion interventions in which people are motivated to change unhealthy behaviour, and (iii) environmental interventions—which are public health interventions focused on modifying the environment such as water purification to prevent water-borne diseases; environmental decontamination and improving air ventilation in enclosed environments to prevent air-borne diseases.

f. *Immune response system*: A type of functionally organized complex system that consists of two sub-systems which are the innate immune response sub-system and the adaptive immune response sub-system: Each of these functionally organized complex sub-systems is organized into three main levels/scales: (i) the molecular level/scale—which includes the proteome, lipodome, genome, metabolome, transcriptome, and complex molecular processes such as gene expression, gene regulatory networks, signalling, and metabolic pathways involved in immunity and inflammation, (ii) the cellular level/scale—which includes the activities and behaviour of the different immune cells such as T-cells, B-cells, and different pathogen processes, and (iii) the tissue level/scale—which includes inflammation processes.

The attractiveness of multiscale modelling of infectious disease systems is that it provides an opportunity to investigate the influence of these functionally organized complex systems on infectious disease dynamics at more than one scale.

4.2. **Use of multiscale models in the analysis of mechanisms of infectious disease dynamics**: This involves using multiscale models to uncover the molecular, cellular, tissue, physiological, etc. mechanisms of infectious disease systems.

4.3. **Use multiscale models in predicting/forecasting of infectious disease dynamics**: This may involve use of multiscale models to forecast/predict when disease risk may be increased or peak or use of multiscale modelling in forecasting/predicting disease burden at a particular level of organization of an infectious disease system.

4.4. **Use of multiscale models in analysis/evaluation/formulation of policy for control or elimination or eradication of an infectious disease system**: This may involve use of multiscale models in planning to provide general guidance on the amount of resources (human, financial, etc.) required to control or eliminate or eradicate an infectious disease system or use of multiscale models to examine the impact of different disease control or elimination or eradication policies. These uses of multiscale models might be for an impending epidemic or a currently unfolding epidemic. Based on [2], and the knowledge of the two different types of levels of organization of an infectious disease system presented in this article (level of multiscale observation and level of multiscale analysis), we can now be able to determine if a multiscale is more appropriate for either the control, or the elimination, or the eradication of an infectious disease system based

on the level of multiscale observation used in the development of the multiscale as follows.

a. *Multiscale models with a primary structure*: These are multiscale models developed using either the cell level or tissue level or the host level as the level of multiscale observation. These multiscale models are more relevant for evaluating the control of infection at specific hierarchical levels of an infectious disease system. Such multiscale models contain a primary multiscale loop/cycle [2] formed by the action of the reciprocal influence between pathogen replication at microscale and pathogen transmission at the macroscale at either the cell level or the tissue level or the host level as the level of multiscale observation. This multiscale loop involves local exchange of pathogen between the microscale and the macroscale of either the cell level or the tissue level or the host level as the level of multiscale observation.

b. *Multiscale models with a secondary structure*: These are multiscale models developed using either the organ level or community level (community level I, community level II, community level III) as the level of multiscale observation. These multiscale models are more relevant for evaluating the elimination of infection at specific hierarchical levels of an infectious disease system. Such multiscale models contain a secondary multiscale loop formed by the action of the reciprocal influence between persistence of infection at the microscale of these levels of multiscale observation (i.e. within-organ scale or within-community scale) and dispersal/spread of infection at the the macroscale of these levels of multiscale observation (i.e. between-organ scale or between-community scale). This secondary multiscale loop involves global exchange of single pathogen species/strain and/or single host species infected with single pathogen species/strain among scales of an infectious disease system.

c. *Multiscale models with a tertiary structure*: These are multiscale models developed using either the microecosystem level (microecosystem level I, microecosystem level II, microecosystem level III) level or the macroecosystem level (macroecosystem level I, macroecosystem level II, macroecosystem level III) as the level of multiscale observation. These multiscale models are more relevant for evaluating the eradication of infection at any hierarchical level of organization of an infectious disease system. Such multiscale models contain a tertiary multiscale loop formed by the action of the reciprocal influence between persistence of infection at the microscale of these levels of multiscale observation (i.e within-microecosystem scale or within-macroecosystem scale) and dispersal/spread of infection at the macroscale of these levels of multiscale observation (i.e. between-microecosystem scale or between-macroecosystem scale). This secondary multiscale loop involves (i) local exchange of multiple pathogen species/strains at microecosystem level (microecosystem level I, microecosystem level II), (ii) global exchange of multiple pathogen species/strains at microecosystem level III, and (iii) global exchange of multiple pathogen species/strains and/or multiple host species infected with multiple/single pathogen species/strains at macroecosystem level (macroecosystem level I, macroecosystem level II, macroecosystem level III).

The formation of multiscale loops/cycles in disease dynamics was recently formalized into a scientific theory for multiscale modelling of infectious disease systems [2]. While some progress has been made in the development of multiscale models of infectious disease systems with a primary multiscale (see [5] and references therein), more research is needed to develop multiscale models with secondary multiscale loops or tertiary multiscale loops.

## Discussion

Multiscale modelling of infectious diseases aims to characterize the complexity of infectious disease systems. It provides us with a set of analytic tools to characterize the multiscale structure of infectious disease systems and how the scales interact with each other through the different multiscale cycles/loops (primary multiscale loop, secondary multiscale loop, tertiary multiscale loop) to establish the observed disease dynamics. The conclusive result of this article is a methodology to design multiscale models of infectious diseases. The methodology is based on implementing a four-stage strategy in the research and development process for multiscale models of infectious disease systems. Further, the methodology is general enough to be applicable (with minor modifications) to multiscale modelling of other structurally organized complex systems beyond infectious disease systems. We eagerly anticipate that this research and development process for multiscale models will support the emergence of multiscale modelling as a branch of complexity science applied to the study of infectious disease systems. While this research and development process for multiscale models cannot be claimed to be unique, complete and final, it constitutes a good starting point, which may be found useful as a basis for further refinement in the discourse for multiscale modelling of infectious disease systems.

## Author Contributions

**Conceptualization:** Winston Garira.

**Methodology:** Winston Garira.

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
