## [Decision Letter · Decision Letter 0]

29 Nov 2019

Dear Dr Garira,

Thank you very much for submitting your manuscript 'The Research and Development Process for Multiscale Models of Infectious Disease Systems' for review by PLOS Computational Biology. Your manuscript has been fully evaluated by the PLOS Computational Biology editorial team and in this case also by independent peer reviewers. The reviewers appreciated the attention to an important problem, but raised some substantial concerns about the manuscript as it currently stands. While your manuscript cannot be accepted in its present form, we are willing to consider a revised version in which the issues raised by the reviewers have been adequately addressed. We cannot, of course, promise publication at that time.

Sincerely,

Jennifer A. Flegg

Guest Editor

PLOS Computational Biology

Virginia Pitzer

Deputy Editor

PLOS Computational Biology

[LINK]

Reviewer's Responses to Questions

**Comments to the Authors:**

Reviewer #1: This manuscript proposes a general framework for developing and using multi scale models of infectious disease systems, to address the lack of standardisation in the way that these types of models have been developed and used to date. This has been submitted as a research article, however it does not fit the usual format of such an article, i.e., introduction which describes a research question and context of the work, results, discussion, methods. It may be more suited to being a review (although there is limited literature cited here beyond the authors own work) or an eduction piece.

Major Comments

1. It is not made clear how this framework relates to those the author has previously proposed in references [1,2]. There are a different number of levels of organisation proposed in each of these works. Why is the update /extension necessary?

2. The proposed framework incorporating sub-systems/levels/scales is complex. The author has provided a number of schematic diagrams to aide understanding. However, I think linking back to biological examples of the boundaries and interactions within/between these different levels of organisation would also be helpful, along with, where appropriate, citing examples in the literature of exemplar multi-scale models and their research questions.

3. It is not clear that the pathogen subsystem proposed is able to encapsulate processes of horizontal gene transfer, e.g., via phage or plasmids, which can occur between pathogen species that occupy the same niche/host and between strains of the same species during co-infection. This is especially important for the spread of antibiotic resistance in pathogen populations, and for vaccine escape, such as that which occurred for Pneumococcus (see Corander et al., 2017: https://doi.org/10.1038/s41559-017-0337-x)

4. It is known that households provide the opportunity for prolonged close mixing between individuals and it is generally assumed that the risk of infection transmission between household contacts far exceeds that in the wider community. Also many interventions target households. Yet, households do not appear in the framework. This should at least be discussed, or incorporated into the framework.

5. Section 2.6: Levels and scales are also linked by exchange of phage, and other mobile genetic elements, not just the pathogen, hosts, host products, etc. This should at least be discussed, or incorporated into the framework.

6. Related to comment 2 above: points a-e, page 22. Reference to exemplar research questions and models would aide understanding of these reasons for model selection

7. Stage III. Point 3.3, page 29. There are two different, but related, types of analyses that should be conducted for any ID model: sensitivity analysis (which parameters are the outputs most sensitive too? i.e., using partial rank correlation coefficient), and uncertainty analysis (how does uncertainty in inputs affect uncertainty in outputs? i.e., using Latin Hypercube sampling of parameter space). This should be acknowledged.

Minor comments

1. The author uses the terms “actual scale” and “characteristic scale” throughout, which I assume mean the same thing. It would be more clear to use one of these terms consistently, especially give ghere is considerable terminology being introduced in the paper around scales/levels/sub-systems.

2. Point (a) on page 3. “the inability to distinguish between local infections and imported infections in the … is because of using single scale modelling instead of multi scale modelling”. Isn’t this a data collection and analysis issue rather than a modelling issue? Such errors can be overcome by collected the right data, e.g., genomic data of isolates, or meta-data associated with isolates

3. Point (c) on page 5. The use of the term “local interactions” is misleading here as it suggests interactions only within a subsystem. Consider using just “interactions”

4. Figure captions should include all details necessary to understand this figure. E.g., what do (i)-(vii) refer to in Figure 1, what do arrows mean? what do (a)-(f) refer to in Figure 4

5. Page 10: throughout points (a)-(d) you use the phrase “is clear”. It is not clear. Please just say what the boundary is.

6. Page 11: model categories I-V need to be described. Also provide examples from literature.

7. Use of the term “super-infection” throughout. When this term is first used it should be defined, as there is not consensus in literature. It is used to mean both strain replacement (superseding infection) as well as co-infection (co-existence) of strains within a host.

8. Page 17: Is it not true that global exchange of organisms can also play a role in disease persistence? E.g, pathogen moving to population with higher levels of susceptibility, and being re-introduced after a period of time, or TB bacteria moving from airways to granuloma where it can persist for long periods of time?

9. Point e, page 31. How do mass drug administrations fit in here? They are treatments, but given to people not necessarily infected.

Reviewer #2: The authors make good case for multiscale modeling of infectious diseases from a complex systems perspective. The paper provides developmental process maps for multiscale modeling systems, in particular the integration of four different multiscaling approaches to address the infectious disease dynamics is interesting.

Despite some interesting ideas, the reviewer finds that the manuscript needs some additional work to be published in a high impact journal.

(1) In the vast literature of epidemic modeling examples of each aspect of multiscale nodeling system proposed by the author.

correlating the existing studies for a well studied epidemic (e.g. AIDS, H1N1, Ebola etc) into the hierarchical format suggested by the author in figure 2 would help identify where the specialized studies fit in a complex systems approach.

(2) The same comment for the integration of multiscale modeling approaches in figure 4. It would be helpful to identify examples of past research that fits into each category. This would be needed to bring out the interplay between different modeling approaches for the suggested integrated system.

**Have all data underlying the figures and results presented in the manuscript been provided?**

Reviewer #1: Yes

Reviewer #2: Yes

PLOS authors have the option to publish the peer review history of their article (what does this mean?). If published, this will include your full peer review and any attached files.

Reviewer #1: No

Reviewer #2: No

---

## [Decision Letter · Decision Letter 1]

13 Feb 2020

Dear Prof. Garira,

We are pleased to inform you that your manuscript 'The Research and Development Process for Multiscale Models of Infectious Disease Systems' has been provisionally accepted for publication in PLOS Computational Biology.

Before your manuscript can be formally accepted you will need to complete some formatting changes, which you will receive in a follow up email. A member of our team will be in touch within two working days with a set of requests.

Best regards,

Jennifer A. Flegg

Guest Editor

PLOS Computational Biology

Virginia Pitzer

Deputy Editor

PLOS Computational Biology

Reviewer's Responses to Questions

**Comments to the Authors:**

Reviewer #1: In the revised manuscript the author has addressed my main concerns. A couple of minor comments:

- I think the concept of super-infection needs only to be defined once on page 17, not three times, also remove from page 35

- Include references for methods listed in lines 1445, 1447. Some suggestions: https://doi.org/10.1098/rsif.2012.1018, https://doi.org/10.2307/1403510

Reviewer #2: The authors have addressed the reviewers comments adequately.

**Have all data underlying the figures and results presented in the manuscript been provided?**

Reviewer #1: Yes

Reviewer #2: Yes

PLOS authors have the option to publish the peer review history of their article (what does this mean?). If published, this will include your full peer review and any attached files.

Reviewer #1: No

Reviewer #2: No

---

## [Editor Report · Acceptance letter]

12 Mar 2020

PCOMPBIOL-D-19-01717R1 

The Research and Development Process for Multiscale Models of Infectious Disease Systems

Dear Dr Garira,

I am pleased to inform you that your manuscript has been formally accepted for publication in PLOS Computational Biology. Your manuscript is now with our production department and you will be notified of the publication date in due course.

With kind regards,

Laura Mallard
